# Sensitivity study of the instrumental temperature corrections on Brewer total ozone column measurements

Alberto Berjón[1,2], Alberto Redondas[3,2], Meelis-Mait Sildoja[4], Saulius Nevas[4], Keith Wilson[5], Sergio F. León-Luis[3,2], Omar el Gawhary[6], and Ilias Fountoulakis[7]

[1]University of La Laguna, Department of Industrial Engineering, S.C. de Tenerife, Spain
[2]Regional Brewer Calibration Center for Europe, Izaña Atmospheric Research Center, Tenerife, Spain
[3]Agencia Estatal de Meteorología, Izaña Atmospheric Research Center, Spain
[4]Physikalisch-Technische Bundesanstalt (PTB), Braunschweig, Germany
[5]Kipp & Zonen BV, Delft, Netherlands
[6]Dutch Metrology Institute, Delft, Netherlands
[7]Aristotle University of Thessaloniki, Laboratory of Atmospheric Physics, Thessaloniki, Greece

*Correspondence to:* Alberto Redondas (aredondasm@aemet.es)

**Abstract.** The instrumental temperature corrections to be applied to the ozone measurements by the Brewer spectrophotometers are derived from the irradiance measurements of internal halogen lamps in the instruments. These characterizations of the Brewer spectrophotometers can be carried out within a thermal chamber, varying the temperature from -5$°C$ to +45$°C$, or during field measurements, making use of the natural change in ambient temperature. However, the internal light source used to determine the thermal sensitivity of the instrument could be affected in both methods by the temperature variations as well, which may affect the determination of the temperature coefficients. In order to validate the standard procedures for determining Brewer's temperature coefficients, two independent experiments using both external light sources and the internal halogen lamps have been performed within the ATMOZ Project. The results clearly show that the traditional methodology based on the internal halogen lamps is not sensitive to the temperature-caused changes in the spectrum of the internal light source. The three methodologies yielded equivalents results, with differences in total ozone column below 0.08% for a mean diurnal temperature variation of 10$°C$.

## 1 Introduction

The Brewer spectrophotometer has been used for decades as a reference instrument to retrieve total ozone column (TOC) and for the validation of satellite-based measurements. TOC is retrieved from direct sun measurements at four wavelengths in the ultraviolet (UV) spectral range, from 310.1 nm to 320.1 nm. To be able to make such measurements, this instrument should be suitable for outdoor use. Moreover, it should operate at any temperature, since it is installed in a wide range of environments, from subtropical deserts to polar zones. Changes in the ambient temperature may affect the Brewer instrument in two different ways: change its spectral responsivity and cause a wavelength shift.

Preventing errors in the wavelength selection is of highest importance for the ozone determination. Therefore, different strategies had been already implemented to minimize this problem and the impact of temperature on the spectral shift is

expected to be minimal. The spectrophotometer contains a temperature-compensated monochromator that allows accurate measurements in the UV range (McElroy, 2014). Materials used in the monochromator are selected to minimize the effect of the internal temperature changes on the spectrum position relative to the exit slits. Nevertheless, mechanical tolerances in the manufacturing may cause imperfections in this temperature compensation. Thus, the Brewer operational procedure

recommends to perform an internal Hg-lamp test (HG test) when the internal temperature, which is registered for each ozone measurement by a sensor located near the PMT, varies more than $3°C$ (Environment Canada, 2008). The HG test uses a mercury discharge lamp (line 302.15 nm or 296.73 nm) to check the stability of the wavelength calibration during Brewer instrument operations. During the test, the diffraction grating is positioned such that the operating wavelengths are dispersed onto the appropriate exit slits (Kipp & Zonen, 2008). Finally, wavelengths used to determine the TOC were selected at stationary points

in the solar spectrum in order to minimize the effect of any residual spectrum shift (Brewer, 1973).

The Brewer spectrophotometer quality assurance protocol includes the thermal characterization of the instrument. This characterization is initially carried out inside a thermal chamber at Kipp & Zonen, manufacturer of the Brewer spectrophotometer, by measuring the output of an internal halogen lamp while varying the temperature of the chamber from -5$°C$ to +45$°C$ during a period of 72 hours. Temperature coefficients are determined during this characterization at all Brewer operating wavelengths

using a linear approximation.

If an appreciable temperature dependence in the retrieved ozone is detected during the calibration campaigns, the temperature coefficients are adjusted using the in-field data of the internal halogen lamp measurements along the diurnal temperature variation (Redondas and Rodríguez-Franco, 2015). A drawback of this method is the narrower temperature range that is available for the field measurements compared with the thermal chamber method. However, the natural ambient temperature variation

applies to the instrument during its normal operation mode, which generally yields acceptable temperature coefficients.

The determination of the thermal sensitivity of the instrument by means of the internal halogen lamp, used in both described procedures, implies that the internal lamp itself and the power supply circuit are also subjected to temperature changes and that they can also exhibit some temperature sensitivity. This can potentially modify the lamp irradiance or its alignment, hampering the determination of the temperature coefficients. In addition, there are different elements involved in the direct sun

measurements but not in the measurement of the internal halogen lamp, such as the quartz window and the neutral density filters, which can make the results of the characterization to be ineffective when applied to the operational measurements.

While the temperature effect on the global UV irradiance measurements by the Brewer spectrometer has been studied by different authors (Weatherhead et al., 2001; Siani et al., 2003; Fountoulakis et al., 2017), so far no validation of the temperature sensitivity of the TOC retrieved from the Brewer measurements has been reported. On this basis, a validation of the standard

procedures used for the retrieval of the temperature coefficients was included as one of the objectives within the European Metrology Research Programme (EMRP) project "Traceability for atmospheric total column ozone" (ATMOZ).

In this work, we report on a comparative study of the temperature coefficients retrieved by the standard procedures (using internal halogen lamps in a thermal chamber and during field measurements) and by using two alternatives setups employing external lamps for thermal chamber measurements. For this purpose, we have made measurements with #185 and #233 MKIII

Brewer spectrophotometers at PTB (Physikalisch-Technische Bundesanstalt) in Braunschweig, Germany, and Kipp&Zonen

in Delft, Netherlands, respectively. For comparison purpose, also field data from the EUBREWNET (COST Action ES1207) database is used (Rimmer et al., 2018).

## 2   Principle of measurement of the Brewer spectrophotometers

In order to understand how the temperature correction is applied to the measurement data by the Brewer spectrophotometers we review in this section the basic measurement principles that are used to obtain the TOC from this instrument. A comprehensive description of the Brewer instrument and the TOC calculation from the results of the measurements can be found in Kerr et al. (1985).

TOC is the main product derived from direct solar irradiance measurements by the Brewer spectrophotometers. The direct irradiance measurement is performed by pointing the direct entrance port normally to the sun based on an azimuth tracker and a rotating quartz prism which follows the sun's elevation. These measurements are made through a quartz window covering the direct irradiance port. To select the different wavelengths used in the TOC derivation, the Brewer spectrophotometer maintains a fixed position of the diffraction grating and uses a rotating slit mask to select successively each wavelength. The rapid movement of the slit mask assures that all wavelengths are measured almost simultaneously. TOC, in Dobson units (DU) or milli-atm-cm, is obtained from Equation 1 (following manufacturer's nomenclature):

$$TOC = \frac{ETC - R_6 - B}{A\mu} \tag{1}$$

where $R_6$ is usually defined on the basis of double ratios of measured intensities, $I_c(\lambda_i)$, at certain wavelengths (Kipp & Zonen, 2008), but it can be also written as the linear combination of the common logarithm of the intensity:

$$R_6 = \sum_{i=1}^{4} \omega_i F_c(\lambda_i) \tag{2}$$

$$F_c(\lambda_i) = 10^4 log(I_c(\lambda_i)) \tag{3}$$

The coefficients, $\omega_i$, take values of -1.0, 0.5, 2.2 and -1.7 for the wavelength of 310.1 nm, 313.5 nm, 316.8 nm and 320.1 nm, respectively. These wavelengths correspond to slits 2 to 5 in the rotating slit mask of the monochromator. $I_c(\lambda_i)$ are obtained from the Brewer raw signal counts after dark count, dead time, temperature and filter transmittance corrections. The wavelengths, $\lambda_i$, used in Equations 2 and 3 have been specially selected to minimize effects of any small shift in the wavelength (Fioletov et al., 2005). Moreover, coefficients $\omega_i$ have been determined to suppress any influence of the aerosol and the $SO_2$

on the ozone retrieval (Dobson, 1957; Kerr et al., 1981). In general, any linear effects with wavelength are also suppressed, as $\lambda_i$ and $\omega_i$ satisfy the conditions defined by Equation 4 and 5.

$$\sum_{i=1}^{4} \omega_i = 0 \tag{4}$$

$$\sum_{i=1}^{4} \omega_i \lambda_i \approx 0 \tag{5}$$

5     $ETC$ is a linear combination (Equation 6) of the extraterrestrial constants of the instrument, $F_{ext}(\lambda_i)$, which can be obtained from a comparison with a calibrated instrument or from the Langley plot method (Redondas, 2007).

$$ETC = \sum_{i=1}^{4} \omega_i F_{ext}(\lambda_i) \tag{6}$$

$B$ is a linear combination of the Rayleigh transmittances of the air, $\beta(\lambda_i)$, corrected by the Rayleigh air mass, $\nu$, and the ratio of the pressure at the observation position, $p$, and the standard pressure at sea level, $p_0$.

10     $$B = \nu \frac{p}{p_0} \sum_{i=1}^{4} \omega_i \beta(\lambda_i) \tag{7}$$

$A$ is a linear combination of the ozone absorption coefficients, $\alpha(\lambda_i)$.

$$A = \sum_{i=1}^{4} \omega_i \alpha(\lambda_i) \tag{8}$$

Both the Rayleigh air mass, $\nu$, and the ozone air mass, $\mu$, are calculated assuming an effective altitude of 5 km and 22 km, respectively (Bernhard et al., 2005).

15     $R_6$ is derived from the intensity ratios or equivalently from Equation 2 and, therefore, it has no units, just like $ETC$ and $B$.

## 3   Temperature correction

Most photodetectors have some sensitivity to temperature. If the sensitivity can be linearly approximated, the intensity $I\ [c/s]$ of a stable light source measured by the detector at different temperatures $T\ [°C]$ can be expressed as:

$$I = I_c - \tau_0(T - T_0) \tag{9}$$

,

where $T_0$ is the reference temperature, $I_c$ is the intensity of the source measured at the reference temperature, and $\tau_0$ is the variation rate of the intensity with the temperature $[c/s°C]$. We can rewrite this expression as:

$$I_c = \frac{I}{1 - \tau(T - T_0)} \tag{10}$$

where $\tau = \tau_0/Ic$ is the temperature coefficient having units of $1/°C$. This last expression has an advantage that, while $\tau_0$ depends on the intensity of the light source, $\tau$ is independent and we can use it to determine the intensity of the source at the reference temperature. This process is generally referred to as temperature correction.

The temperature coefficient is usually determined in a laboratory measuring a stable light source while the detector temperature is varied. $\tau$ is calculated from the linear regression between measured intensity and the temperature. From the previous equation and applying the natural logarithm, we can write:

$$ln(I_c) = ln(I) - ln(1 - \tau(T - T_0)) \tag{11}$$

For Brewer spectrophotometers $\tau \approx 10^{-3}\,°C^{-1}$, then $\tau(T-T_0) \ll 1$ and we can approximate the natural logarithm $ln(1+x)$ to the first order term of its Taylor expansion:

$$ln(I_c) = ln(I) + \tau(T - T_0) \tag{12}$$

In the Brewer data processing, this expression is multiplied by $10^4$, the natural logarithm is replaced by common logarithm and $T_0$ is set to $0°C$.

$$10^4 log(I_c) = 10^4 log(I) + \tau_b T \tag{13}$$

Where $\tau_b = 10^4 log(e)\tau$ is the Brewer temperature coefficient. Using Equation 3 we can rewrite this expression as:

$$F_c = F + \tau_b T \tag{14}$$

We can define a Brewer relative temperature coefficient, $\tau_b'(\lambda_i)$, by subtracting the reference coefficient from coefficients derived for other spectral channels. Usually the reference coefficient is the one corresponding to the wavelength $\lambda_0 = 303.2nm$. In terms of the Brewer temperature coefficient we can express the relative coefficient as:

$$\tau_b'(\lambda_i) = \tau_b(\lambda_i) - \tau_b(\lambda_0) \tag{15}$$

Since the weights used in the common Brewer algorithm to calculate ozone are chosen to satisfy the Equation 4, we can rewrite the Equation 2 as:

$$R_6 = \sum_{i=1}^{4} \omega_i F_c(\lambda_i) = \sum_{i=1}^{4} \omega_i F(\lambda_i) + T \sum_{i=1}^{4} \omega_i \tau_b(\lambda_i) = \sum_{i=1}^{4} \omega_i F(\lambda_i) + T \sum_{i=1}^{4} \omega_i \tau_b'(\lambda_i) \tag{16}$$

Equation 16 shows that we can use interchangeably $\tau_b(\lambda_i)$ or $\tau_b'(\lambda_i)$ to calculate $R_6$ and, therefore, TOC.

5    The relative temperature coefficients, $\tau_b'(\lambda_i)$, can be calculated from Equation 15, but they can also be experimentally retrieved by the linear regression between the ratio of intensities, expressed as $F(\lambda_i) - F(\lambda_0)$, and the temperature:

$$F(\lambda_i) = F_c(\lambda_i) - \tau_b(\lambda_i)T$$
$$F(\lambda_i) - F(\lambda_0) = F_c(\lambda_i) - F_c(\lambda_0) - (\tau_b(\lambda_i) - \tau_b(\lambda_0))T$$
$$F(\lambda_i) - F(\lambda_0) = F_c(\lambda_i) - F_c(\lambda_0) - \tau_b'(\lambda_i)T \tag{17}$$

The relative coefficients have the advantage that they can be reliably derived even if the illumination condition is not stable. The only requirement for the determination of the relative coefficients is that the change of the light source is proportional at 10    all wavelengths.

Furthermore, from the previous Equations we can see that the temperature effect over $R_6$ can be reduced to the linear combination of the temperature coefficients, $\tau_{R6}$.

$$\tau_{R6} = \sum_{i=1}^{4} \omega_i \tau_b(\lambda_i) \tag{18}$$

Where $\tau_b(\lambda_i)$ can refer indistinctly to the Brewer temperature coefficients or to the relative temperature coefficients. $\tau_{R6}$ is 15    expected to be more robust to changes in the light source than the relative coefficients $\tau_b'(\lambda_i)$. Since $w_i$ satisfy the Equation 5, $\tau_{R6}$ can be correctly determined not only if the change of the light source is proportional at all wavelengths, but also if the change is linear with the wavelength.

## 4    EUBREWNET data on operating temperatures and thermal sensitivities

In order to determine the most suitable temperature range for the experiments, a statistical analysis of the Brewer operating 20    temperatures using the EUBREWNET database was performed. The standard operating ambient temperature range provided by the manufacturer of the Brewer spectrophotometer is from $0°C$ to $+40°C$. These range limits are related to the operating temperature range of the photomultiplier tube (PMT), which is from $0°C$ to $+50°C$. As the heat dissipated by the internal electronics increases the temperature in the instrument by about $5°C$, the operating temperature range of the Brewer has a safety margin of $5°C$.

In the case that the ambient temperature drops below $0°C$, the equipment should be operated using an internal heater that allows to extend the operating ambient temperature range down to -20$°C$. In practice, the internal heater is activated when instrument temperature drops below $10°C$ or $20°C$ depending of the instrument configuration. Additionally, a cold-weather cover is furnished by the manufacturer for extreme weather conditions, which allows operating the instrument at ambient temperatures as low as -50$°C$.

From the EUBREWNET database (http://rbcce.aemet.es/eubrewnet) we have studied the Brewer spectrophotometer internal temperatures at which the ozone measurements have been made in 32 measurement stations worldwide from 1996 to 2017. For this analysis, 4.2 million recorded temperatures were used. The stations involved in the study are mainly in Europe, but there are also data from Greenland, Australia, Uruguay and Algeria. They are, therefore, a very representative sample of the different environmental conditions under which the Brewer spectrophotometers are measuring throughout the world. Figure 1 shows a boxplot for all the stations. The median temperature values for all the stations are between $16°C$ and $32°C$, while the 1st and 3rd quartiles are always above $11°C$ and below $39°C$, respectively. The mean diurnal temperature variation is $12°C$. Figure 2 shows a histogram with data from all stations together. Considering all the data, the mean temperature value is $23.0°C$ and a median value is $22°C$, with a standard deviation of $16.6°C$. The 1 and 99 percentiles are estimated to correspond to $5°C$ and $44°C$. Only a small number of measurements (0.04%) are outside the safety limits for the PMT that we discussed earlier ($0°C$, $50°C$).

We have also analyzed the thermal sensitivity, $\tau_{R6}$, of 44 Brewer spectrophotometers included in the EUBREWNET database, Figure 3. These values range from -0.9$°C^{-1}$ to 4.0$°C^{-1}$. Two different distributions appear to be clearly related to the different Brewer models. MKIII model has a mean $\tau_{R6}$ value of 0.20$°C^{-1}$ and a standard deviation of 0.52$°C^{-1}$. In the case of MKII and MKIV, the mean value rises to 1.54$°C^{-1}$ and the standard deviation is 0.70$°C^{-1}$. While Brewer MKIII has a double monochromator to assure a low stray light influence in UV, MKII and MKIV have a single monochromator, so that they use a $NiSO_4$/UG11 filter in front of the PMT to eliminate the effect of visible light on the measurements, i.e., to reduce the stray light in the UV range. The higher temperature dependence observed in the single-monochromator Brewers is commonly attributed to this $NiSO_4$/UG11 filter (Fountoulakis et al., 2017; Cappellani and Kochler, 2000).

## 5 Experimental setups

To study the temperature sensitivity of the Brewer spectrophotometers by using external lamps, two different experimental setups have been used: a first one at PTB in Braunschweig, Germany and a second one at Kipp & Zonen in Delft, Netherlands (both shown in Figure 4). For these studies, MKIII Brewers #185 and #233 were chosen. These instruments are the traveling master instrument of the Regional Brewer Calibration Center for Europe (RBCC-E) triad at the Izaña observatory of the Spanish National Meteorological Agency (AEMET) and a research instrument of Kipp & Zonen, respectively.

During the experiments the internal heater was turned off, but the air circulation fan was left on to evenly distribute the air inside the Brewer instrument, allowing a uniform heating up and cooling down of the internal components.

For the Brewer #185 characterization, a dedicated climate chamber at PTB was used to provide the necessary conditions. The schematic of the measurement system is presented in Figure 5 (left). The temperature and humidity of the chamber was monitored using the built-in sensors of the chamber and two extra sensors, one PT-100 thermometer and one Almemo humidity and temperature sensor. A Hamamatsu model LC8 UV source with a built-in $Xe$ lamp and equipped with a quartz
fiber bundle as a light guide was used to illuminate simultaneously both the global and the direct input ports of the Brewer spectrophotometer. The light guide was terminated with light-shaping-diffuser (LSD) to provide a uniform illumination. To monitor the output stability of the UV source, a set of monitor detectors were placed close to the Brewer input ports. Those included two $SiC$ photodiodes and a entrance optics of a calibrated spectroradiometer, which was outside of the chamber. One of the $SiC$ photodiodes was located inside the chamber and the other one outside. To direct the UV-radiation onto the external
$SiC$ diode a similar light guide was used as for the $Xe$ lamp system. For optimal irradiation conditions the internal $SiC$ photodiode and the entrance optics of the spectroradiometer included special quartz-based Primusil diffusers. For the external $SiC$ photodiode, no diffuser was used to compare the readings with the diffuser-covered detectors and register any possible change of the diffuser transmittance due to the change of the temperature or the relative humidity in the chamber during the experiment. The Brewer observations consisted of alternately measuring the internal and the external lamps. The external $Xe$
lamp was continuously on during the whole cycle of the characterizations while the internal halogen lamp was turned on and off for each measurement. The drift of the $Xe$ source irradiance at the Brewer entrance port was corrected by using the calculated mean of the normalized integrated spectral data from the monitor spectroradiometer and the temperature-corrected $SiC$ detector readings.

The experiment was done twice at the PTB facilities. On the first occasion in January 2016, referred in the results as PTB1,
the quartz fiber bundle was used to illuminate simultaneously both the global and the direct input ports of the Brewer instrument. The temperature of the climate chamber was varied between -5$°C$ to +40$°C$ over a 70 hours period. Separate cycles were used above and below 0$°C$ to achieve a better control over the temperature and the humidity. Due to some inconsistencies in the retrieved results of the first experiment, the temperature characterizations were repeated at the PTB facilities in February 2017. On this second occasion, referred to as PTB2, the light guide of the external lamp was aligned to illuminate only the direct
entrance port. The global port was not used for the measurements. In addition, the internal halogen lamp was replaced since anomalous behavior was observed during its operation at the time of the first measurements in January 2016 and later also at the RBCC-E. Two different temperatures cycles were used at different temperature change rates. First, the temperature of the climate chamber was varied between 8$°C$ and 45$°C$ over a 64 hours period with a temperature change rate of 1.2$°C/h$. A second cycle was done varying the temperature from -8$°C$ to 30$°C$ over 50 hours with a temperature change rate of 2.8$°C/h$.
The experimental setup for characterizing Brewer #233 is shown in Figure 5 (right). The temperature in the chamber was varied from -5 to +45 during a period of 72 hours. A Laser Driven Light Source (LDLS) Energetiq EQ-99, from Dutch Metrology Institute (VSL), was used as an external lamp. By means of an optical fibre bundle, the light was guided into the chamber, collimated by a 25 mm lens and then illuminated the Brewer's quartz window at normal incidence. The collimated beam illuminated the rotating prism, which was aligned in accordance with the incoming light. During the external lamp
measurements, no lamp monitor was used, as one of the main characteristics of the LDLS is its high stability (Islam et al.,

2013). The other components in the beam delivery part were assumed to be stable and independent of temperature. Two separate experiments were performed using the internal halogen lamp in the first case and the external LDLS via the quartz window in the second case. The respective lamps were continuously turned on during each experiments.

In all cases, each measurement cycle included an HG test, repositioning of the micrometer of the diffraction grating to locate the 302.15 nm line of the mercury discharge lamp.

Since different instruments have been used in each experiment (#185 at PTB and #233 at K&Z) the differences in results may be due not only to the differences of the experiment setup, but also due to the different Brewer instruments.

In this work we also include an analysis of the measurements based on the internal halogen lamp during field measurements at dates close to the characterizations in the temperature chambers.

## 6 Results

Results from the experiments carried out at PTB and at Kipp & Zonen, as well as analysis of the field measurements are presented in this section.

In Figure 6 we show results of the Brewer measurements of both the external sources and the internal halogen lamps at different temperatures. For the sake of brevity, only the data for 310.1 nm wavelength are shown. The measurements at the other wavelengths show a very similar behavior. The first evident thing apparent in the figures is the difficulty of assuming a linear behavior in these measurements. Despite of using different experimental setups and instruments, the results are not as expected. Only the relation between the measurement results of the internal halogen lamp and the temperature in the K&Z experiment can be considered linear. Nevertheless, some behaviors are repeated in the two experiments at PTB, which makes us assume that they are probably due to real changes in the behavior of lamps, detectors or the different mechanical elements during the experiments. One of these clearly observed behaviors is the presence of hysteresis cycles. This is possibly related to an inhomogeneous temperature distribution within the instrument. In any case, it is difficult to extract information from the data presented in this way.

The total variation shown in the plots in Figure 6 is only about 1% for a temperature change of more than $50°C$ over nearly three days. Under these conditions it is difficult to ensure sufficient stability of the light sources and the required precision in the alignment. Any uncontrolled effect during the measurements may negatively impact the determination of the absolute temperature coefficients. These results reflect the difficulty of determining the absolute temperature coefficients for the Brewer spectrophotometer.

As stated in section 3, the relative coefficients are intended to be more robust against variations of the experimental conditions. Figures 7, 8 and 9 show results of the relative analysis of the measurements presented versus temperature and the relative temperature coefficients derived from both the external and the internal lamps in the three experiments. The analysis of the internal halogen lamp data from field measurements is also presented.

We can clearly see the improvement when using the analysis of the relative coefficients. In most of the cases we can assume a linear relationship between relative Brewer measurements and the temperatures. Figures 7, 8 and 9 show two different linear

regressions for each data sets. In a first approximation we use the individual relative Brewer measurements. In a second step, a linear regression is performed with respect to the average values at each temperature. This is done to avoid overrepresentation of any of the temperatures since the number of measurements at certain temperatures is much higher than at the others. A summary of the results obtained by both linear regressions methods is shown in Tables 1 and 2.

The analysis of the internal halogen lamp measurements in PTB1 shows a very marked nonlinear behavior when using data from slits 4 and 5 relative to measurements with slit 1. Due to this, only data below 30°C have been used to make the linear regression to obtain the relative temperature coefficients. This behavior was not repeated in PTB2. This change in the behavior must have been due to the replacement of the internal halogen lamp between both experiments. One week before the PTB1 experiment, the internal halogen lamp was burned out and it had to be replaced. However, this replaced lamp used in the PTB1

experiment did not show a stable behavior also during later field measurements. Therefore, it was replaced again in March 2016. This is also the reason why we have only a few number of measurement points to make the field data analysis shown in Figure 7. Moreover, differences between external lamp measurements in PTB1 and PTB2 may be due to the alignment changes to couple the external lamp radiation into the direct entrance port. All these problems makes us to consider Brewer #185 not stable during the PTB1 experiment and, therefore, its results will not be included in the final analysis.

Fortunately, the PTB2 experiment (Figure 8) showed more consistent results. Here, the relationship between the relative measurement results and the temperature are mainly linear, although some non-linearities are observed for both the internal and the external lamp measurements between 40°C and 50°C. This non-linearity for high temperatures may be explained by the photomultiplier tube behavior at these temperatures. Figure 10 shows the dark signals for Brewers #185 and #233 measured during PTB2 and K&Z experiments. They demonstrate different behaviors in the operating temperature range. Moreover, at

temperatures between $40°C$ and $50°C$ both show an increase in the dark signals. Additionally, dark signals of #185 show two different trends between $40°C$ and 50°C along the whole experiment time. From the data sets of the EUBREWENET data base we can see that the most usual behavior corresponds to the one shown by #233. The K&Z experiment gives also a linear relation between the relative data and the temperature for both the internal and the external lamp measurements, Figure 9.

     From Tables 1 and 2 we can see that the determined relative temperature coefficients present important differences depending

on the used data set (thermal chamber measurements with internal halogen lamp and external lamp or field measurements with internal lamp) in both PTB2 and K&Z experiments. However, when calculating $\tau_{R6}$ the differences are strongly reduced and very similar values are obtained, as shown in Figure 11.

     Differences between the $\tau_{R6}$ values retrieved from the internal and the external lamp measurements are less than 0.06 for both PTB2 and K&Z experiments, independently if the linear regression is done using individual values or mean values at each

temperature. Including the field data retrieval, differences are within about 0.08 in both PTB2 and K&Z experiments when the linear regression is done using mean values at each temperature. For the regression using individual values, difference rise to 0.14 and 0.07 for the PTB2 and K&Z data sets, respectively. The higher coefficient found in the case of the PTB2 field data set when the linear regression is done using individual values, indicates that the greatest number of measurements taken at lower temperatures has a negative impact on the estimated coefficient.

The typical value of the numerator in Equation 1, $ETC - R_6 - B$, is about 1000 for an optical airmass of 1, thus differences in $\tau_{R6}$ of 0.08 represent a 0.08% of TOC for a temperature variation of $10°C$. As the mean diurnal variation is close to $10°C$, that value can be considered the diurnal uncertainty due to the temperature correction. This result might be useful when analyzing different operating issues, such as wrong temperature coefficients or incorrect values of $ETC$, that may introduce
diurnal cycles in the final ozone values.

Note that the uncertainty associated to the different coefficients in Tables 1 and 2 correspond to the standard uncertainty of the slopes from the linear regressions in Figures 7, 8, 9 and 11. $\tau_{R6}$ coefficients can also be calculated as a linear combination of the relative coefficients or directly from the linear regressions in Figure 11. However, to derive the associated uncertainty of $\tau_{R6}$ from the uncertainties of the relative coefficients we should assume they are not independent variables. Therefore, the
calculation of $\tau_{R6}$ uncertainty is more direct from the linear regressions.

It is worth to note that temperature correction is usually applied to measurement data using a reference temperature close to the most frequently faced operation temperature. A reference temperature close to the mean operational temperature means that the applied temperature correction is most of the time small and, thus, a low accuracy estimation of the temperature sensitivity will not have a high effect on the TOC retrieval. However, this is not the case with the Brewer spectrophotometer, which
use a reference temperature of $0°C$ while the mean operation temperature is $23°C$ and a median value of $22°C$. Considering the negligible uncertainty of the temperature measurements, we can write the uncertainty associated with the temperature correction as $\delta\tau_{R6}(T-T_0)$, where $\delta\tau_{R6}$ is the uncertainty associated to the coefficient $\tau_{R6}$. Figure 12 shows an estimation of the uncertainties of the temperature corrections applied to the measurements in EUBREWNET database using the distribution of temperatures previously shown in Figure 2 and the mean value of $\delta\tau_{R6} = 0.08°C^{-1}$ that we have obtained from the differences
between the methods. As we can see, the estimated uncertainties are close to zero for most of the cases when $T_0 = 22°C$ and the maximum uncertainty would only be half the maximum uncertainty in the case of $T_0 = 0°C$. While a change of temperature reference form $T_0 = 0°C$ to $T_0 = 22°C$ may increase the uncertainty associated with some measurements made at low temperatures, in general it would result in a reduction of the uncertainty.

## 7   Conclusions

Two experiments were conducted at PTB (in January 2016 and in February 2017) and the Kipp & Zonen (in October 2016) facilities to validate the standard methods for the determination of the temperature dependence of measurements by the Brewer MKIII instruments used to retrieve atmospheric TOC. We have prioritized the MKIII model in this study since it is the most extended model and generally used as reference, as in the case of the RBCC-E.

The first experiment performed at PTB in January 2016 with an unstable internal lamp in the Brewer spectrophotometer
led to unusable results but it showed that it is necessary to guarantee a good performance of the Brewer instrument before carrying out the temperature sensitivity analysis. This highlights the importance of the method based on the internal halogen lamp measurement data in the field since it presents the best way to ensure the correct functioning of the spectrophotometer

throughout its operation. In order to apply this method it is advisable to schedule SL tests throughout the day to record as wide a range of temperature variation as possible.

The absolute temperature coefficients obtained through the different methods present important inconsistencies that prevent their use. These problems are probably due to the difficulties to control the whole system with the required precision. The total variation of the measurement results is only about 1% with a temperature change of more than $50°C$. Uncontrolled effects during the measurements may negatively impact the determination of the absolute temperature coefficients.

The relative results seem to be more robust against uncontrolled systematic effects and they present an approximately linear behavior with the temperature. However, the derived relative temperature coefficients show important differences depending on the data used for their calculation.

The calculated $\tau_{R6}$ values are very much stable. The combination of the four wavelengths clearly increase the stability in all the experiments. This is probably because the linear combination removes any linear effect with wavelength, as it satisfies the conditions defined by Equation 5. Instead, the relative coefficients does not satisfy this property.

Better results are found when the linear regression is done using the mean value at each temperature. The TOC differences due to the method used to calculate the temperature coefficients $\tau_{R6}$ are below 0.08% for a mean diurnal temperature variation of $10°C$.

These experiments confirm that the characterizations performed in a thermal chamber using either the internal halogen lamp of the Brewer instrument or an external lamp as well as those carried out with the internal halogen lamp during field measurements lead to small differences in the retrieved $\tau_{R6}$. This is the case even though the values of the relative coefficients are obtained using different types of lamps.

The analysis of the EUBREWNET data shows some temperature sensitivity differences between Brewer MKIII model and the MKII and MKIV models, which may be related to $NiSO_4$/UG11 filter. Therefore, the conclusions of this work may not be directly applicable to the MKII and MKIV models. Further studies are necessary in order to analyze these specific models.

Finally, it is advisable to consider a change of the reference temperature from the actual $0°C$ to $22°C$. While this could increase the uncertainty associated with some measurements made at low temperatures, in general it would result in a reduction of the uncertainty associated with the temperature correction.

*Competing interests.* The authors declare that they have no conflict of interests.

*Acknowledgements.* This work has been supported by the European Metrology Research Programme (EMRP) within the joint research project ENV59 "Traceability for atmospheric total column ozone" (ATMOZ). The EMRP is jointly funded by the EMRP participating countries within EURAMET and the European Union. We thank the European Brewer Network (http://rbcce.aemet.es/eubrewnet/) for providing access to the data and the PI investigators and their staff for establishing and maintaining the 32 sites used in this investigation. We further acknowledge the support of the Fundación General de la Universidad de La Laguna.

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

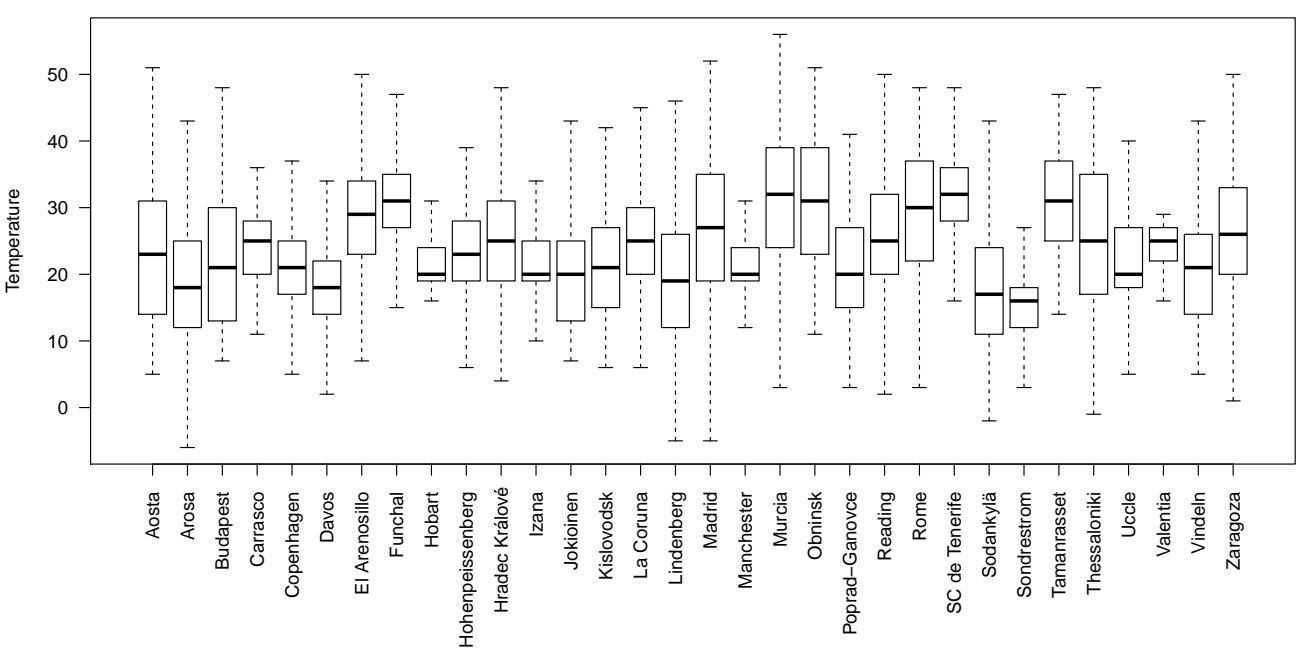

**Figure 1.** Statistics of each of the 32 stations used for the analysis of the operational instrumental temperatures. Median temperatures are represented by lines within boxes determined by the 1st and 3rd quartiles. Whiskers represent Tukey's limits. The 1st and 3rd quartiles are always above $11°C$ and below $39°C$ respectively.

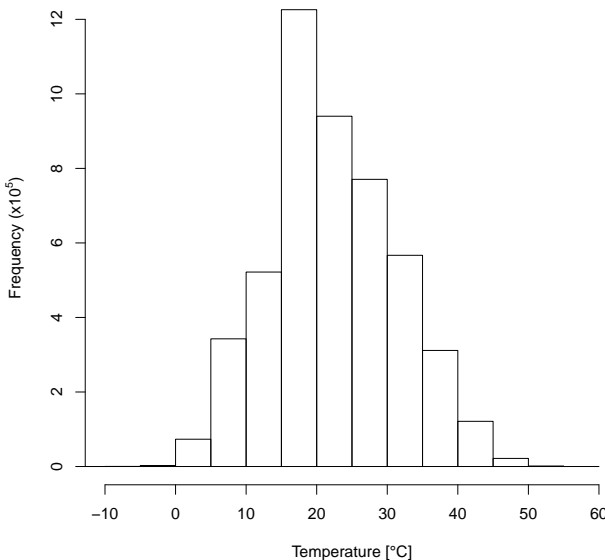

**Figure 2.** Frequency distribution of instrument temperatures from EUBREWNET database. 4.2 million temperature data points have been used. Only a 0.04% of measurements are outside the recommended limits ($0^\circ C$ and $50^\circ C$).

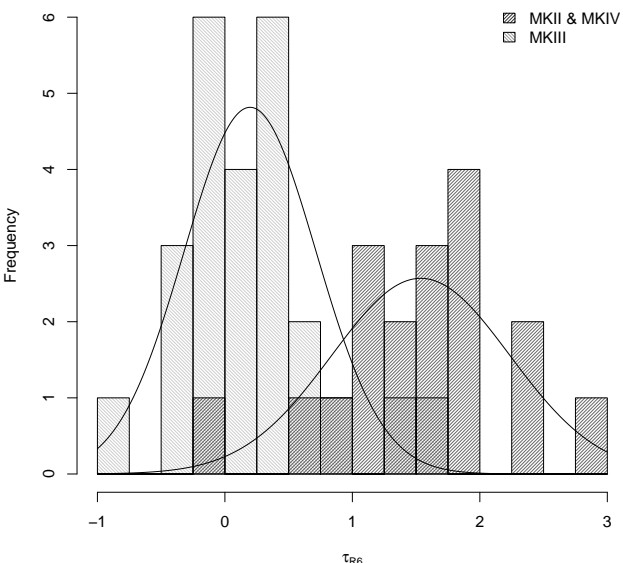

**Figure 3.** Temperature sensitivity of the different Brewer models.

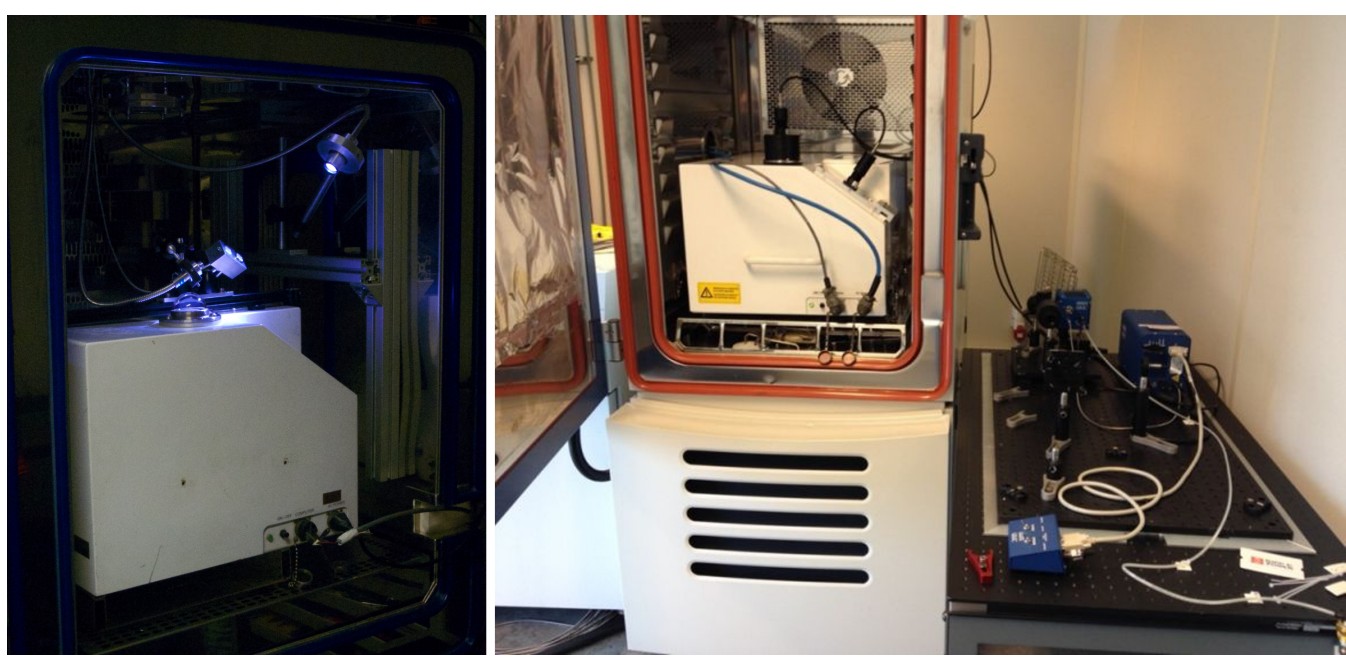

**Figure 4.** Pictures showing Brewer instruments in the controlled environment chambers for the measurements at the PTB (left) and at the Kipp & Zonen facilities (right).

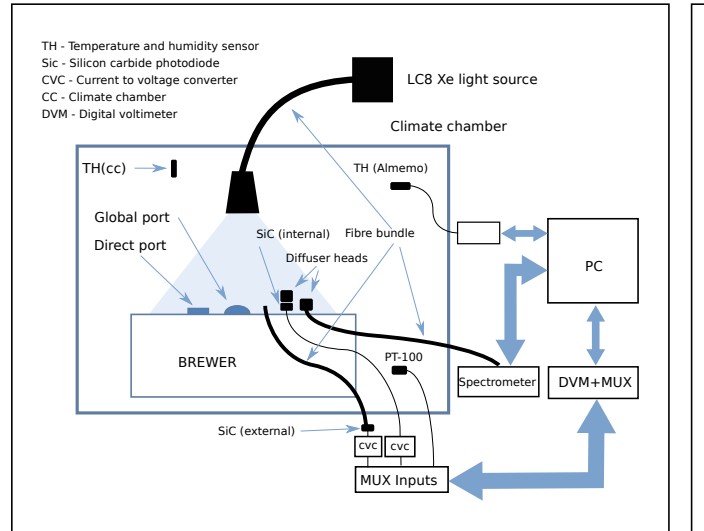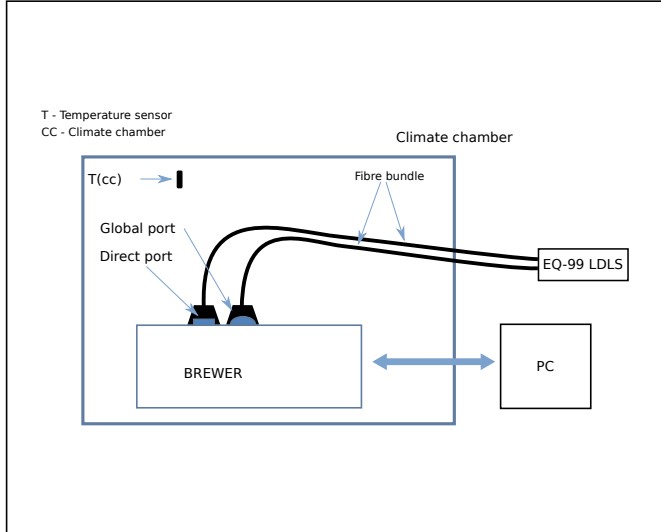

**Figure 5.** Measurement setups for used for the Brewer characterizations at PTB (left) and at Kipp & Zonen (right).

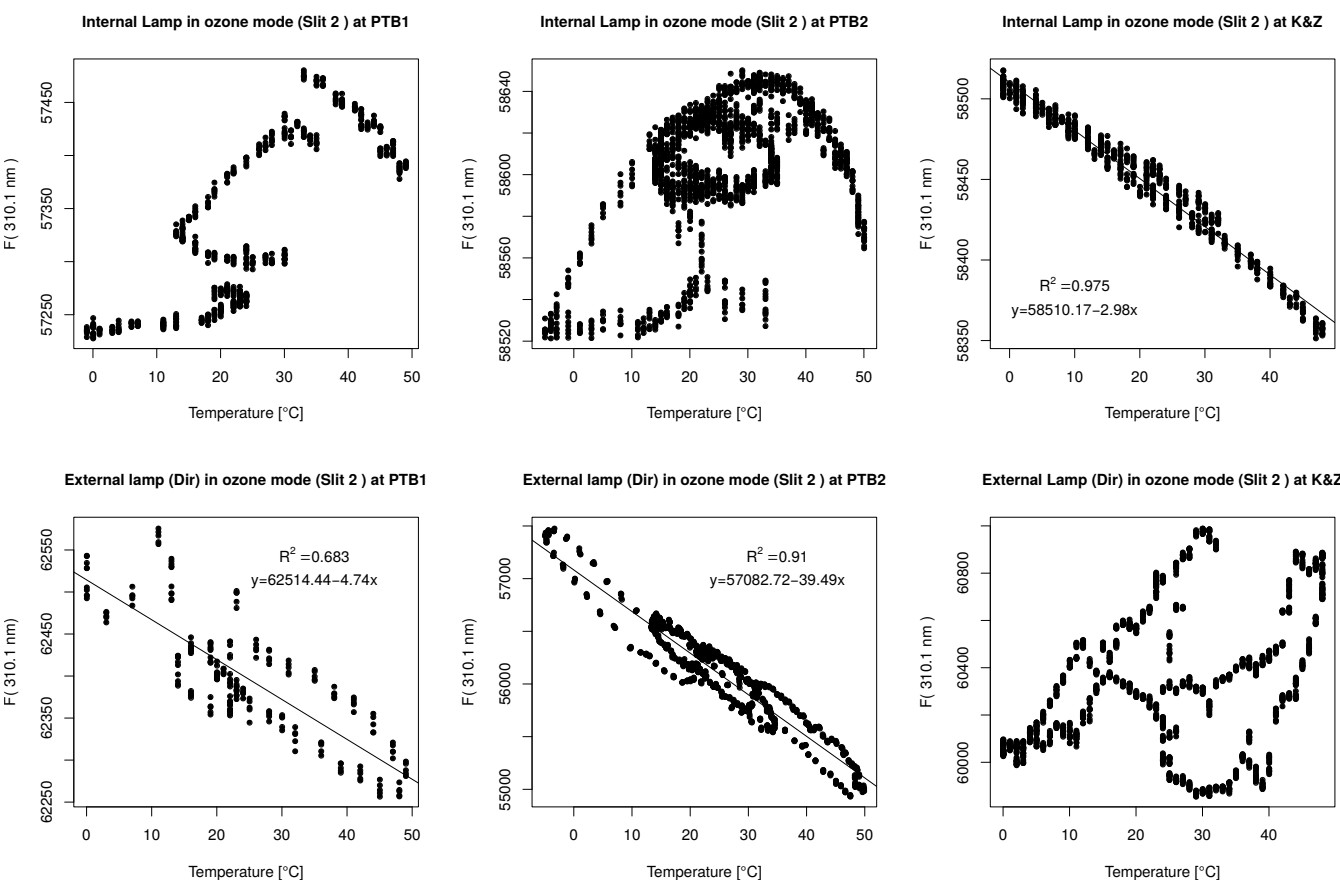

**Figure 6.** Scatter plot of Brewer measurement data (Slit 2) versus temperature in the temperature chambers from PTB1, PTB2 and K&Z experiments when illuminating with the internal (upper plots) and the external lamps (lower plots).

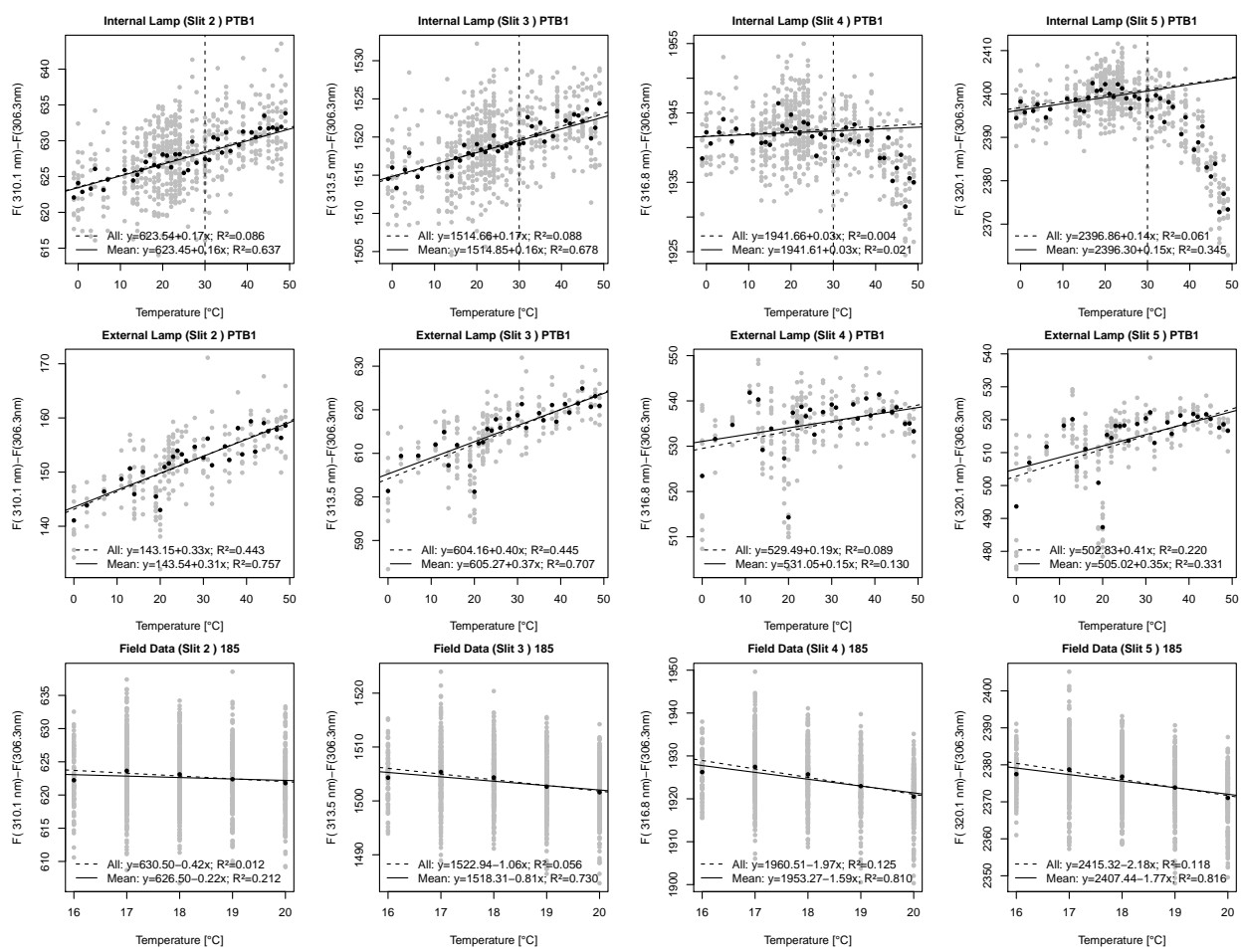

**Figure 7.** PTB1 experiment. Linear regressions between relative Brewer data (Slit 2, 3, 4 and 5 relative to Slit 1) and temperature. Coefficients are determined from internal and external lamp measurements in the temperature chamber and from field data. Linear regression using individual measurements (grey points) is represented by dashed lines, while solid lines represent the linear regression when using average values at each temperature.

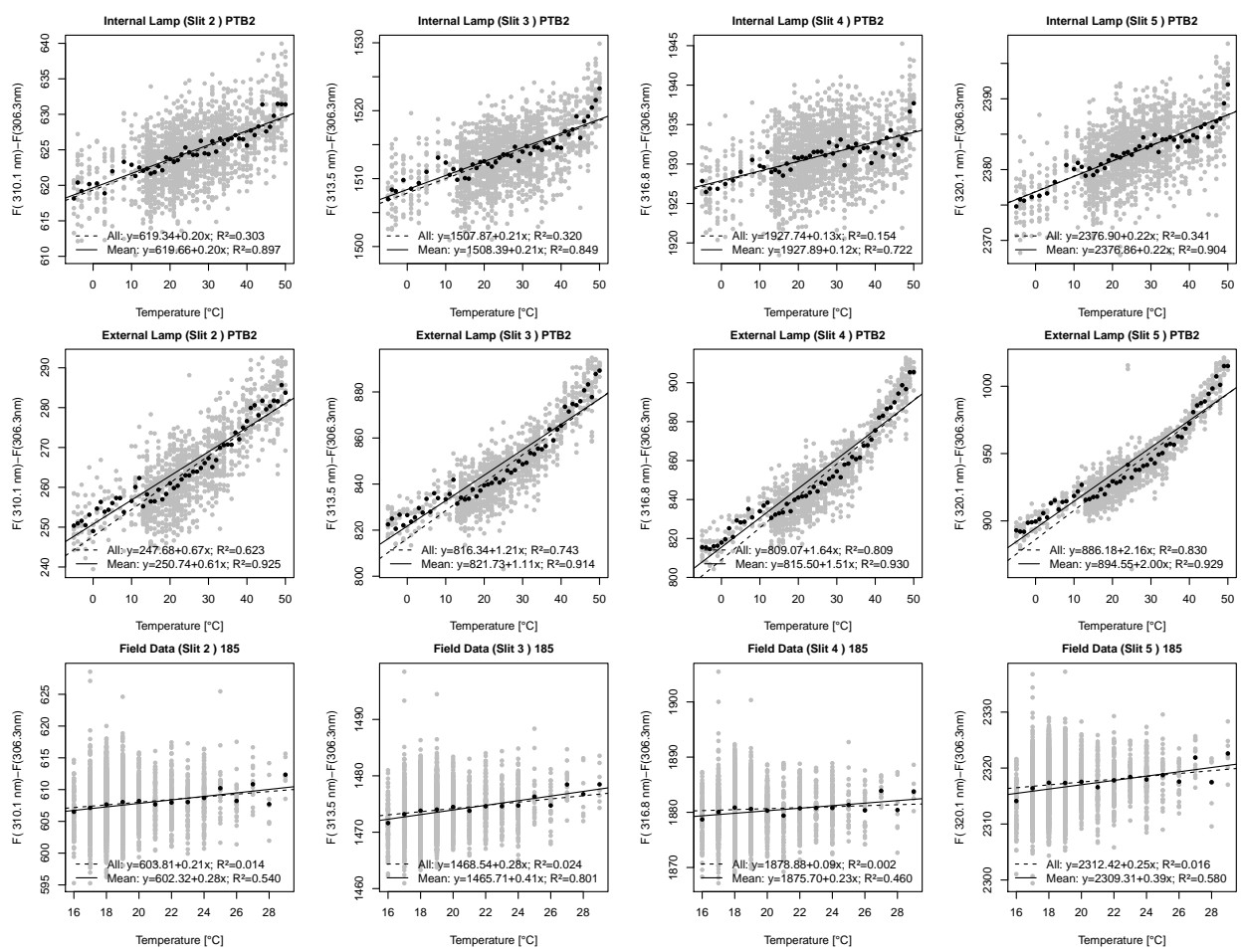

**Figure 8.** PTB2 experiment. Linear regression between relative Brewer data (Slit 2, 3, 4 and 5 relative to Slit 1) and temperature. Coefficients are determined from internal and external lamp measurements in the temperature chamber and from field data. Linear regression using individual measurements (grey points) is represented by dashed lines, while solid lines represent the linear regression when using average values at each temperature.

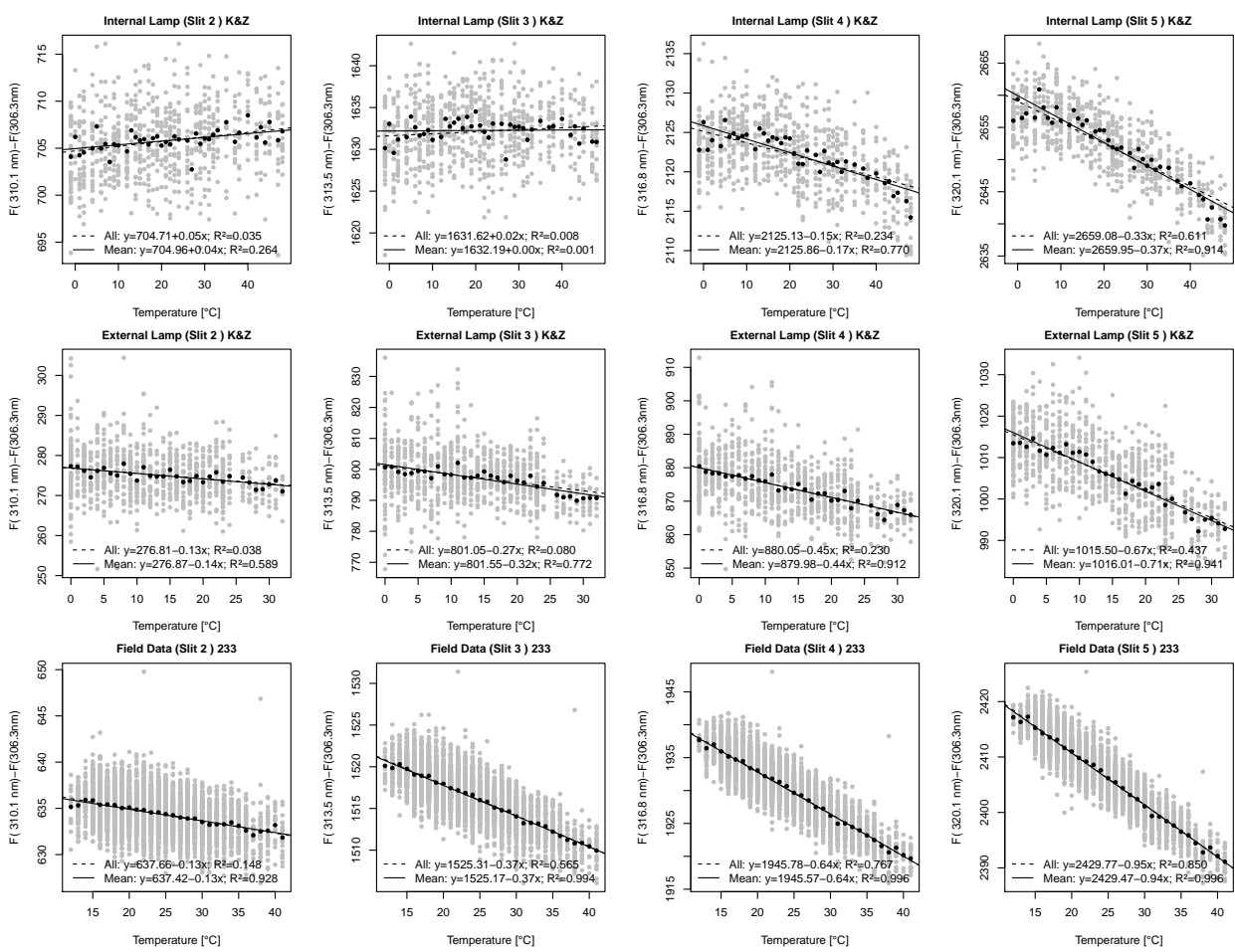

**Figure 9.** K&Z experiment. Linear regression between relative Brewer data (Slit 2, 3, 4 and 5 relative to Slit 1) and temperature. Coefficients are determined from internal and external lamp measurements in the temperature chamber and from field data. Linear regression using individual measurements (grey points) is represented by dashed lines, while solid lines represent the linear regression when using average values at each temperature.

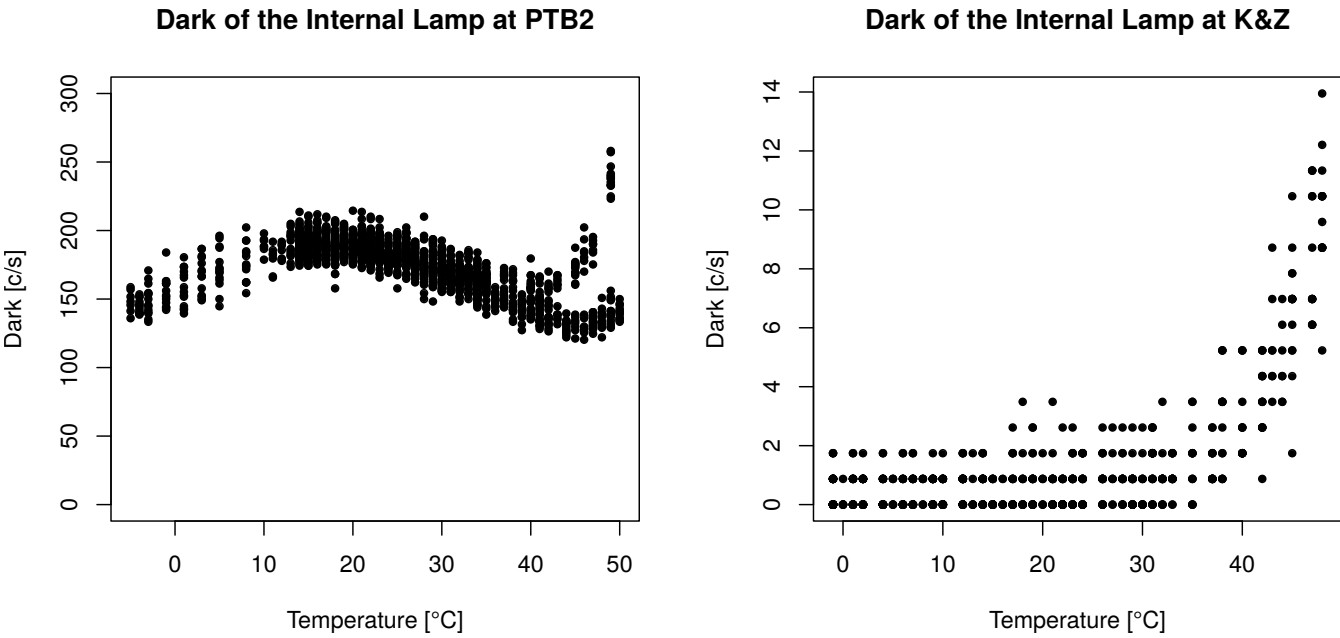

**Figure 10.** Dark signal values during the PTB2 and the K&Z experiments.

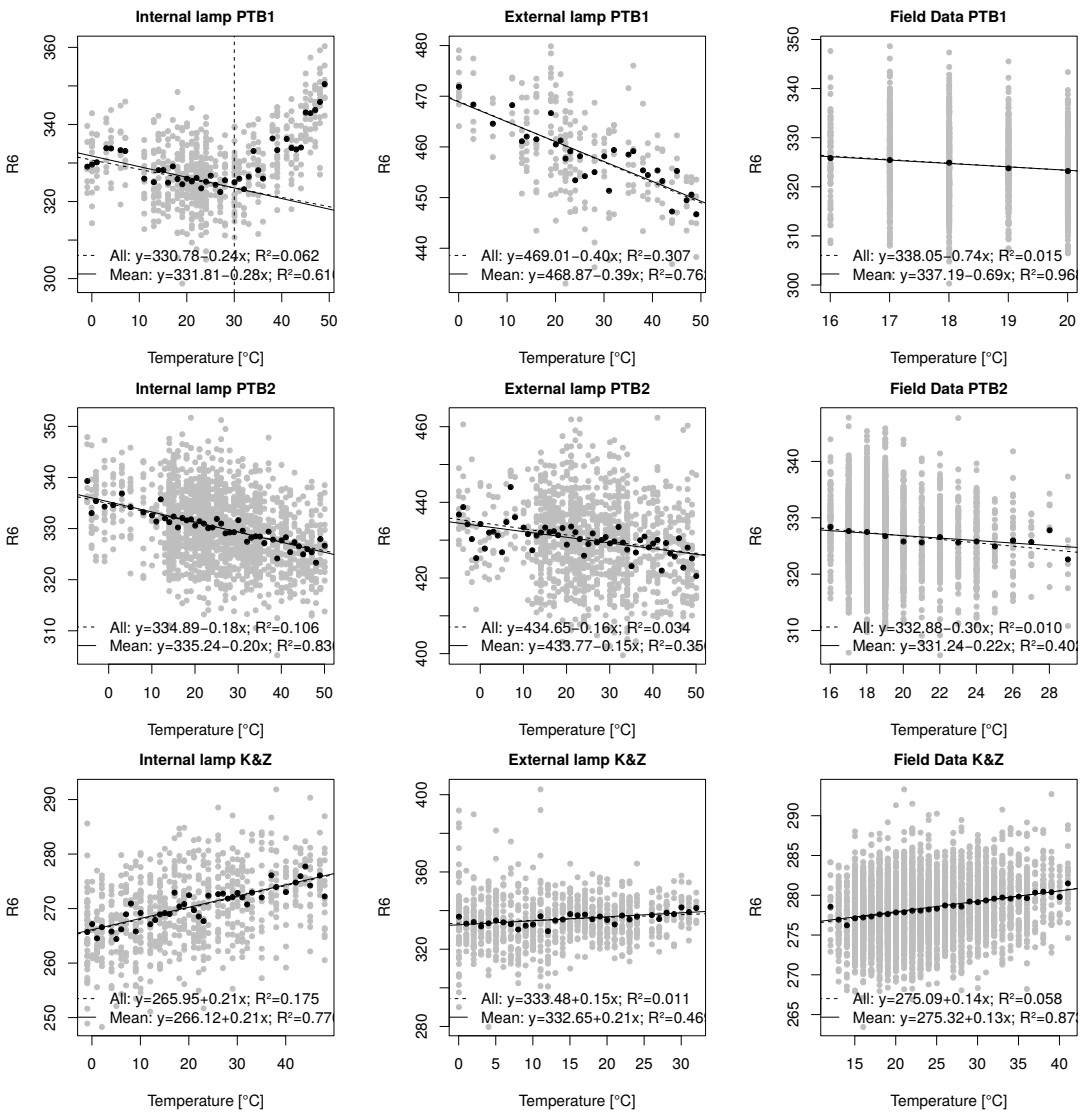

**Figure 11.** Linear regressions between $\tau_{R6}$ and temperature for PTB1, PTB2 and K&Z experiments using internal and external lamp in the temperature test chamber and from field data. Linear regression using individual measurements (grey points) is represented by dashed lines, while solid lines represent the linear regression when using average values at each temperature.

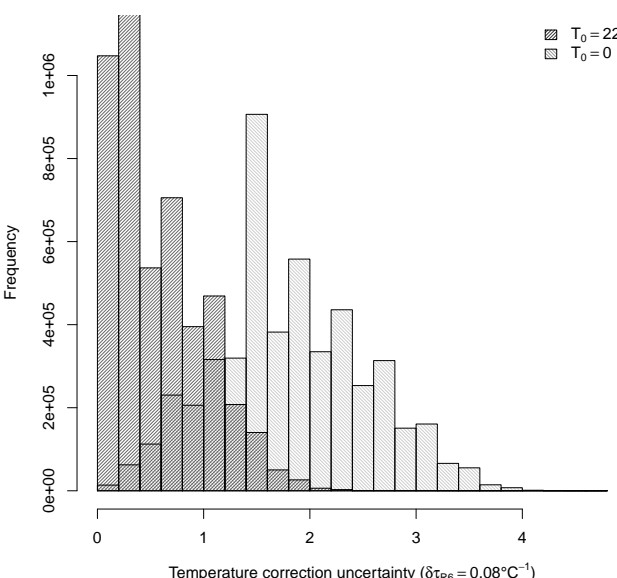

**Figure 12.** Distribution of the temperature correction uncertainties using $\delta\tau_{R6} = 0.08^\circ C^{-1}$ for the reference temperature $T_0 = 0^\circ C$ and $T_0 = 22^\circ C$.

**Table 1.** Temperature coefficients obtained from the linear regression of the individual measurements, for slit 2, 3, 4 and 5 retrieved from Brewer measurements relative to slit 1 ($\tau_b'$) and temperature coefficient for $R_6$ ($\tau_{R6}$). Rows are grouped in three blocks representing the tree experiments (PTB1, PTB2 and K&Z). For each experiment, results from external (top row) and internal (middle row) lamp measurements in the temperature chamber, and derived from field data (bottom row) are shown.

| Experiment | $\tau_b'(310nm)$ $[^\circ C^{-1}]$ | $\tau_b'(313nm)$ $[^\circ C^{-1}]$ | $\tau_b'(316nm)$ $[^\circ C^{-1}]$ | $\tau_b'(320nm)$ $[^\circ C^{-1}]$ | $\tau_{R6}$ $[^\circ C^{-1}]$ |
|---|---|---|---|---|---|
| PTB1 Ext.Lamp | $-0.33 \pm 0.02$ | $-0.40 \pm 0.03$ | $-0.19 \pm 0.04$ | $-0.41 \pm 0.05$ | $0.40 \pm 0.04$ |
| PTB1 Int.Lamp | $-0.17 \pm 0.03$ | $-0.17 \pm 0.03$ | $-0.03 \pm 0.03$ | $-0.14 \pm 0.03$ | $0.24 \pm 0.05$ |
| PTB1 Field | $0.42 \pm 0.11$ | $1.06 \pm 0.12$ | $1.97 \pm 0.15$ | $2.18 \pm 0.18$ | $0.74 \pm 0.18$ |
| PTB2 Ext.Lamp | $-0.67 \pm 0.02$ | $-1.21 \pm 0.02$ | $-1.64 \pm 0.02$ | $-2.16 \pm 0.03$ | $0.16 \pm 0.03$ |
| PTB2 Int.Lamp | $-0.20 \pm 0.01$ | $-0.21 \pm 0.01$ | $-0.13 \pm 0.01$ | $-0.22 \pm 0.01$ | $0.18 \pm 0.01$ |
| PTB2 Field | $-0.21 \pm 0.03$ | $-0.28 \pm 0.03$ | $-0.09 \pm 0.03$ | $-0.25 \pm 0.03$ | $0.30 \pm 0.05$ |
| K&Z Ext.Lamp | $0.13 \pm 0.02$ | $0.27 \pm 0.03$ | $0.45 \pm 0.03$ | $0.67 \pm 0.03$ | $-0.15 \pm 0.05$ |
| K&Z Int.Lamp | $-0.05 \pm 0.01$ | $-0.02 \pm 0.01$ | $0.15 \pm 0.01$ | $0.33 \pm 0.01$ | $-0.21 \pm 0.02$ |
| K&Z Field | $0.13 \pm 0.01$ | $0.37 \pm 0.01$ | $0.64 \pm 0.01$ | $0.95 \pm 0.01$ | $-0.14 \pm 0.01$ |

**Table 2.** Temperature coefficients obtained from the linear regression of the mean values for each temperature, for slit 2, 3, 4 and 5 retrieved from Brewer measurements relative to slit 1 ($\tau_b'$) and temperature coefficient for $R_6$ ($\tau_{R6}$). Rows are grouped in three blocks representing the tree experiments (PTB1, PTB2 and K&Z). For each experiment, results from external (top row) and internal (middle row) lamp measurements in the temperature chamber, and derived from field data (bottom row) are shown.

| Experiment | $\tau_b'(310nm)$ $[^\circ C^{-1}]$ | $\tau_b'(313nm)$ $[^\circ C^{-1}]$ | $\tau_b'(316nm)$ $[^\circ C^{-1}]$ | $\tau_b'(320nm)$ $[^\circ C^{-1}]$ | $\tau_{R6}$ $[^\circ C^{-1}]$ |
|---|---|---|---|---|---|
| PTB1 Ext.Lamp | $-0.31 \pm 0.03$ | $-0.37 \pm 0.05$ | $-0.15 \pm 0.07$ | $-0.35 \pm 0.09$ | $0.39 \pm 0.04$ |
| PTB1 Int.Lamp | $-0.16 \pm 0.03$ | $-0.16 \pm 0.02$ | $-0.03 \pm 0.04$ | $-0.15 \pm 0.04$ | $0.28 \pm 0.05$ |
| PTB1 Field | $0.22 \pm 0.24$ | $0.81 \pm 0.29$ | $1.59 \pm 0.45$ | $1.77 \pm 0.49$ | $0.69 \pm 0.07$ |
| PTB2 Ext.Lamp | $-0.61 \pm 0.02$ | $-1.11 \pm 0.05$ | $-1.51 \pm 0.06$ | $-2.00 \pm 0.08$ | $0.15 \pm 0.03$ |
| PTB2 Int.Lamp | $-0.20 \pm 0.01$ | $-0.21 \pm 0.01$ | $-0.12 \pm 0.01$ | $-0.22 \pm 0.01$ | $0.20 \pm 0.01$ |
| PTB2 Field | $-0.28 \pm 0.07$ | $-0.41 \pm 0.06$ | $-0.23 \pm 0.07$ | $-0.39 \pm 0.09$ | $0.22 \pm 0.08$ |
| K&Z Ext.Lamp | $0.14 \pm 0.02$ | $0.32 \pm 0.03$ | $0.44 \pm 0.03$ | $0.71 \pm 0.03$ | $-0.21 \pm 0.04$ |
| K&Z Int.Lamp | $-0.04 \pm 0.01$ | $-0.00 \pm 0.01$ | $0.17 \pm 0.01$ | $0.37 \pm 0.02$ | $-0.21 \pm 0.02$ |
| K&Z Field | $0.13 \pm 0.01$ | $0.37 \pm 0.01$ | $0.64 \pm 0.01$ | $0.94 \pm 0.01$ | $-0.13 \pm 0.01$ |