# Peer review of "Sensitivity study of the instrumental temperature corrections on Brewer total ozone column measurements"

_Atmospheric Measurement Techniques, 2017_

## Referee Comment (RC1) · J. Rimmer (Referee) · 29 Nov 2017

This paper describes clearly the importance of temperature correction in the Brewer Spectrophotometer and the methodology employed to apply compensation to the derived total ozone column. Since temperature coefficients are traditionally determined from measurements on the internal lamp, it is important that the associated assumptions are proved valid. The results show that the relative corrections are valid, largely as a result of the nature of the ozone algorithm but further investigation would be necessary before applying absolute corrections to absolute measurements such as aerosol optical depth or UV measurements.

This paper should be published with a few technical corrections:

P1 Line 12: Insert 'a' between as and reference.

P3 Line11: Change obtaining to obtained.

P7 Line 30: Should be 'For these studies'

P9 Line19: Remove 'the' from in front of 'section ' Figures 7, 8: Lower plots are labelled 'Filed Data' rather than 'Field Data'

P4 Line 13: 'Most photo-detectors have' would read better.

Eqn 16 has too many equals signs.

P6 Line 6: Should it not be that the change of the response with T is proportional at all wavelengths?

Also, mentions of EUBREWNET should be referenced to COST Action ES1207

---

## Referee Comment (RC2) · V. Savastiouk (Referee) · 4 Dec 2017

December 4, 2017

**Review of "Characterization of the instrument temperature dependence of Brewer total ozone column measurements" by Alberto Berjón et al.**

Vladimir Savastiouk

The paper explores an important question related to the Brewer spectrophotometer TOC measurements/calculations: can the internal halogen lamp be trusted for establishing the temperature coefficients (TC) for correcting the TOC derivation? The result of this project confirms what has been an established practice - the internal lamp tests are sufficiently representative for the TC calculations.

General comments:

While the paper implies that the results can be generalized to all Brewers it doesn't discuss how the presence of the NiSo4+UG11 filter can affect (or not) the results.

It is extremely important in my opinion to clearly distinguish and address separately the two very different effects that temperature has on the Brewer spectrophotometer measurements: 1) a spectrum shift due to difference in temperature coefficient of expansion between glass and metal and 2) changes in the PMT+order filter combination (if one is present) sensitivity/transmission (potentially wavelengths dependent).

A more important comment is about the terminology for the experiments that are described in the paper. The experiments are referred to by the location where they took place, ptb and K&Z, but these two locations/methods were only used for one Brewer each and not for both Brewers mentioned in the paper. This is important when comparing the results: the differences may be due to location/equipment or the Brewer instrument or both.

Also, PTB1 was by all accounts a failed experiment due to poor SL bulb and I do not see any advantage in presenting these results other than mentioning that a stable and reliable SL bulb is needed for TC calculations.

Many scatter plots of R6 vs T show variability for a given temperature greater (often significantly greater) than the difference in R6 at the extremes of the temperature range when using mean values for each temperature. Calculating TC from such data is very questionable as depending on the number of points at each temperature the mean can change if you wait long enough. In other words, TC=0 is all you can say in such situations.

Line-by-line comments:

P2 L4 Strictly speaking, positioning of the grating at the operating wavelength is not part of the HG test, this is done after the test. If you want to be more precise try re-writing this sentence.
P2 L6 "may not be perfect" seems out of place as this is precisely the reason HG test is done. Maybe this sentence was intended to be earlier in the text?

P2 L25 ground quartz filter is used in both DS and SL measurements

P3 L10 maybe it is worth mentioning that the movement of the slit mask is "rapid" to indicate that all wavelengths are measured almost simultaneously.

P3 L18 it is common practice to number slits 0 to 5 with dark count having no slit number (it is not a slit) and numbering slit mask motor positions 0 to 7 with position 1 corresponding to dark count

P3 L19 Rayleigh is also explicitly corrected for

P4 L20 I couldn't quite figure out what the advantage is in having tau/I vs tau, I just hope readers are smarter than me.

P5 EQ16 and all that leads to it - seems like a very complicated way to show that an addition of a constant doesn't change the convolution with weighting coefficients that add to zero by themselves

P6 L5 the requirement is same as my previous comment - constant in log space - seems like a more precise definition than "the change of the light source is proportional at all wavelengths". Please re-phrase.

P7 L32 I happen to know that #233 is now installed in Malaysia and cannot possibly be K&Z reference. Please verify this information or say "it used to be" reference.

P8  Please make clear that #185 was only in PTB and #233 was only at K&Z

P8 L10 There seems to be no results or consequences of having photodiodes. Why mentioning them?

P10 L4 PMT's dark count is theoretically proportional to the exponent of temperature. This is clearly the case with #233, but not with #185. Is it possible that there was something wrong in the setup of #185?

P11 L16 Maybe it is worth explaining that having reference temperature close to median operational temperature means little to know correction due to temperature is needed most of the time and thus there is slightly more room for TC inaccuracies.

---

## Author Comment (AC1) · 6 Feb 2018

We would like to thank Dr. John Rimmer for all his constructive suggestions and comments. They have been quite useful to improve the paper. We include as additional information the .pdf file of the discussion paper with all changes highlighted.

**P1 Line 12: Insert 'a' between as and reference.**

Done.

**P3 Line11: Change obtaining to obtained.**

[Figure]

Done.

**P7 Line 30: Should be 'For these studies'**

Done.

**P9 Line19: Remove 'the' from in front of 'section ' Figures 7, 8: Lower plots are labelled 'Filed Data' rather than 'Field Data'**

Done.

**P4 Line 13: 'Most photo-detectors have' would read better.**

Done.

**Eqn 16 has too many equals signs.**

In order to simplify Eq. 16, the intermediate steps have been deleted.

**P6 Line 6: Should it not be that the change of the response with T is proportional at all Wavelengths?**

In this sentence we refer to the light source intensity. Our intention is to explain how relative coefficients can be successfully calculated even using no very stable measurements, as it can happen when using field data.

**Also, mentions of EUBREWNET should be referenced to COST Action ES1207**

We have include the reference to COST Action ES1207 in the introduction as well as in the acknowledgements.

Please also note the supplement to this comment:
https://www.atmos-meas-tech-discuss.net/amt-2017-406/amt-2017-406-AC1-
supplement.pdf

**Supplement:**

[revised manuscript text omitted]

---

## Author Comment (AC2) · 6 Feb 2018

We would like to thank Dr. Vladimir Savastiouk for all his constructive suggestions and comments. They have been quite useful to improve the paper. We include as additional information the .pdf file of the discussion paper with all changes highlighted.

**General comments:**

**GC#1: While the paper implies that the results can be generalized to all Brewers**

[Figure]

**it doesn't discuss how the presence of the NiSo4+UG11 filter can affect (or not) the results.**

Brewer MKIII model have been used in the experiments carried out in the present study, both in the PTB and in K&Z. In addition, section 4 shows a study about the Brewer Operating temperature in the EUBREWENET stations, which include Brewer models MKII & MKIV. This section is complemented with a stadistic of temperature sensitivity of the different Brewer models (Fig. 3), which show a different behavior between the MKIII model and the MKII & MKIV models. Therefore the conclusions of this work can not be extended to the MKII & MKIV models.

To clarify this point we have modified the following sentence of the conclusions:

*... of the temperature dependence of the Brewer measurements used to retrieve atmospheric TOC.*

By

*... of the temperature dependence of Brewer MKIII measurements used to retrieve atmospheric TOC.*

And included the following sentence:

*The analysis of the EUBREWNET data shows some temperature sensitivity differences between Brewer MKIII model and the MKII and MKIV models, may be related with $NiSO_4$ filter. Therefore the conclusions of this work can not be extended to the MKII and MKIV models.*

**GC#2: It is extremely important in my opinion to clearly distinguish and address separately the two very different effects that temperature has on the Brewer spectrophotometer measurements: 1) a spectrum shift due to difference in temperature coefficient of expansion between glass and metal and 2) changes in the**

**PMT+order filter combination (if one is present) sensitivity/transmission (potentially wavelengths dependent).**

We agree with the referee in the importance of this point. However it is out of the scope of this work and it should be addressed in future works, designing an experiment which focus on this objective.

**GC#3: A more important comment is about the terminology for the experiments that are described in the paper. The experiments are referred to by the location where they took place, ptb and K&Z, but these two locations/methods were only used for one Brewer each and not for both Brewers mentioned in the paper. This is important when comparing the results: the differences may be due to location/equipment or the Brewer instrument or both.**

The used nomenclature is intended to emphasize that the two experiments are different indeed.

We consider also important the possible influence of using two different instruments in both experiments. But using the same instrument is not a guarantee either that the results will not be conditioned by the state of the instrument, as shown by the repetitions PTB1 and PTB2.

**GC#4: Also, PTB1 was by all accounts a failed experiment due to poor SL bulb and I do not see any advantage in presenting these results other than mentioning that a stable and reliable SL bulb is needed for TC calculations.**

We have decided to include the PTB1 measurements due to existence of unexpected hysteresis cycles in the absolute measurements that are ratified in PTB2. We believe it is important to show the two repetitions to clarify that these cycles are not due to problems in the experiment.
**GC#5: Many scatter plots of R6 vs T show variability for a given temperature greater (often significantly greater) than the difference in R6 at the extremes of the temperature range when using mean values for each temperature. Calculating TC from such data is very questionable as depending on the number of points at each temperature the mean can change if you wait long enough. In other words, TC=0 is all you can say in such situations.**

The trends observed in the mean R6 vs T are in general also observed in the non averaged R6.

We have used the R6 average for each temperature because the number of measurements for certain temperatures is much higher than for others, which can lead to an overweighting of certain temperatures. Averaging also reduce the outliers effect in the regressions. In general, the differences between the coefficients obtained after averaging and no averaging is less than the error shown in Table 1.

**Line-by-line comments:**

**P2 L4 Strictly speaking, positioning of the grating at the operating wavelength is not part of the HG test,this is done after the test. If you want to be more precise try re-writing this sentence.**

We have corrected the sentence:

*The HG test uses a mercury discharge lamp to precisely locate the 302.15/296.73 nm mercury line.*

By

*The HG test uses a mercury discharge lamp (line 302.15nm or 296.73 nm) to check*

*the stability of the wavelength calibration during Brewer operations.*

**P2 L6 "may not be perfect" seems out of place as this is precisely the reason HG test is done. Maybe this sentence was intended to be earlier in the text?**

The sentence is relocated in the previous text:

*To avoid this effect, the material of the push rod that controls the movement of the diffraction grating is selected so that its contraction and expansion causes the opposite effect on the spectrum, thus minimizing the effect of the temperature on the measurements. Nevertheless mechanical tolerances in the manufacturing may cause imperfections in this temperature compensation, thus the Brewer operational procedure recommends...*

**P2 L25 ground quartz filter is used in both DS and SL measurements**

The sentence:

*In addition, there are different elements involved in the direct sun measurements but not in the measurement of the internal lamp, such as the quartz window, the ground-quartz diffuser and the neutral density filters,*

Is replaced by:

*In addition, there are different elements involved in the direct sun measurements but not in the measurement of the internal lamp, such as the quartz window and the neutral density filters,*

**P3 L10 maybe it is worth mentioning that the movement of the slit mask is "rapid" to indicate that all wavelengths are measured almost simultaneously.**

The sentence:

*To select the different wavelengths used in the calculation, the Brewer spectrophotometer maintains a fixed position of the diffraction grating and uses a rotating slit mask to select successively each wavelength.*

Is replaced by:

*To select the different wavelengths used in the calculation, the Brewer spectrophotometer maintains a fixed position of the diffraction grating and uses a rotating slit mask to select successively each wavelength. The rapid movement of the slit mask assure that all wavelengths are measured almost simultaneously.*

**P3 L18 it is common practice to number slits 0 to 5 with dark count having no slit number (it is not a slit)and numbering slit mask motor positions 0 to 7 with position 1 corresponding to dark count**

In the present work we have follow the same slit number nomenclature than in previously published works (Redondas et al., 2014). Furthermore, in order to avoid confusion the wavelengths along with the slit number have always been included in the graphs.

**P3 L19 Rayleigh is also explicitly corrected for**

To facilitate the understanding of the formulation, Rayleigh correction has been written separately from instrumental corrections in Eq. 1, as in other previously published works (Redondas et al., 2014). In any case, the temperature correction is always made before the Rayleigh correction, so results are equivalent.

**P4 L20 I couldn't quite figure out what the advantage is in having tau/I vs tau, I just hope readers are smarter than me.**

As detailed in the manuscript, $\tau_0$ is dependent of the measured Intensity. Thus to

perform a temperature correction we need not only $\tau_0$ but also $I_c$, or more briefly $\tau = \tau_0/I_c$.

**P5 EQ16 and all that leads to it - seems like a very complicated way to show that an addition of a constant doesn't change the convolution with weighting coefficients that add to zero by themselves**

The intermediate steps of the equation have been deleted.

**P6 L5 the requirement is same as my previous comment - constant in log space - seems like a more precise definition than "the change of the light source is proportional at all wavelengths". Please re-phrase.**

Both Irradiance and Log spaces are equivalent. However, for the non-specialist readers, the log space can be confusing. Therefore, to simplify the reading, we consider it appropriate to keep the sentence.

**P7 L32 I happen to know that #233 is now installed in Malaysia and cannot possibly be K&Z reference. Please verify this information or say "it used to be" reference.**

We correct this error. #233 is not a reference but a research instrument of K&Z. Thus we change the sentence to:

. . . *and a research instrument of Kipp & Zonen, respectively.*

**P8 Please make clear that 185 was only in PTB and 233 was only at K&Z**

We add the following sentence in Section 5:

Since different instruments have been used in each experiment (#185 at PTB and

**233 at K&Z) the differences in results may be due not only to the differences of the experiment setup, but also to the different Brewer instrument.**

**P8 L10 There seems to be no results or consequences of having photodiodes. Why mentioning them?**

We correct the misunderstanding by adding the sentence:
*The drift of the $Xe$ source irradiance at the Brewer entrance port was corrected by using the calculated mean of the normalized integrated spectral data from the monitor spectroradiometer and the temperature-corrected $SiC$ detector readings.*

**P10 L4 PMT's dark count is theoretically proportional to the exponent of temperature. This is clearly the case with #233, but not with #185. Is it possible that there was something wrong in the setup of #185?**

The possible causes of the effect observed in the dark current are under study at the moment. But it is likely to be related to some parameter in the instrument setup. It is worth noting that after this work this check has been included in the routine RBCC-E control to assure the correct operation of the instrumentation.

**P11 L16 Maybe it is worth explaining that having reference temperature close to median operational temperature means little to know correction due to temperature is needed most of the time and thus there is slightly more room for TC inaccuracies.**

We modify the sentence:

*Finally, it is worth to note that temperature correction is usually applied to measurement data using a reference temperature close to the most frequent operation temperature. However, ... will reduce the uncertainty associated with the uncertainty of the temper-*

*ature correction.*

By:

*Finally, it is worth to note that temperature correction is usually applied to mea-surement data using a reference temperature close to the most frequent operation temperature. A reference temperature close to the mean operational temperature means that the applied temperature correction is most of the time small and thus a low accurate estimation of the temperature sensitivity will not have a high effect over the TOC retrieval. However, ... will reduce the TOC uncertainty associated with the uncertainty of the temperature correction.*

Please also note the supplement to this comment:
https://www.atmos-meas-tech-discuss.net/amt-2017-406/amt-2017-406-AC2-supplement.pdf

**Supplement:**

[revised manuscript text omitted]

---

## Referee Report (RR1)

February 28, 2018

Review of "Characterization of the instrument temperature dependence of Brewer total ozone column measurements" by Alberto Berjón et al.

Vladimir Savastiouk

The revised text of the paper addressed many of the concerns raised in the initial review and it reads much better now. I did find other areas where the paper can be improved, some of them are very important.

1. The title sounds as if the paper investigates the Brewer ozone calculation temperature dependence in general, but the content is mostly a comparison of 2 different experiments on 2 different Brewers plus a mentioning of variability of temperature coefficient values among the Brewers in EUBREWNET. I am not sure any general statements can be definitively drawn from this. A title similar to "An investigation of TC determination in EUBREWNET Brewers" can be more precise

P1 L20 Push rod doesn't really control the movement of the diffraction grating. Suggest not going into this details and say "materials used in the monochromator are selected to minimize the effect of the internal temperature changes on the spectrum position relative to the exit slits."

I will repeat my comment in the initial review: the paper does not clearly differentiate between two completely different temperature effects in the Brewer: the positioning of the spectrum and the changes in the spectral sensitivity. Some paragraphs refer to both effects that is extremely confusing. Moreover, the paper doesn't actually investigate anything relating to the spectral shifts and so it is unclear why the authors even go into this area. A simple statement that is in page 1 is more than enough and I feel that no mentioning of this should be anywhere in the paper after that.

P2 L5 "During the test, the diffraction grating is positioned such that the operating wavelengths are dispersed onto the appropriate exit slits" - please rephrase as this sentence doesn't describe much.

P2 L14 TC are calculated for all operating wavelengths, not just those used for ozone

P2 L25 replace "the" with "some" in "quartz window and the neutral density filters"

P3 L7 remove double "an"

P3 L15 it's not really an "alternative", it's the same

P3 L19 I refuse to accept your explanation that since you've already used this notation for the slit numbering in another paper and this makes it ok to use it again. It is wrong to use unconventional terminology, Many papers before yours used it correctly.

P6 L10-18  Should not be in this paper.

P7 L18  Have you looked at the fact that Brewers have at least three different PMT types and that can contribute to the differences in the TC? If you did, what was the result? If not, why not?

P7 L22,25 The order filter in not just $NiSO_4$ crystal, but a combination filter with two UG11 glass filters.

P10 L20 *** An important point!!! *** : did you calculate the slopes and their uncertainties using the averages for each temperature? It very much looks like you did and then your uncertainty is incorrect as you forgot that each point (each average) has an uncertainty associated with it already (a very large standard deviation in fact).  You should re-calculate those uncertainties using each point or propagate the uncertainties of the averages to that of the slopes. When you do this you will likely find that the uncertainty is larger than 100% for most of your slopes, which brings back my original point from the initial review: if  for each temperature you have a spread of R6 so large that is close to the spread between R6 at extreme temperatures you cannot actually correct or improve such data. Whatever you do you will still have that spread. So, the question then is can you even trust such a data for determining TC?

P11 L1 I am not sure I understand why you are saying "in spite of robustness of the TOC calculation algorithm". The algorithm clearly works. It was the instrument (the hardware) that didn't.

 P11 L5 You may consider stating clearly that it is imperative to schedule SL tests throughout the day to cover the different temperatures inside the Brewer.

---

## Referee Report (RR2)

GENERAL COMMENTS

This paper addresses an important issue for the Brewer users and ozone communities, since an accurate assessment of the Brewer temperature dependence is essential to ensure reliable TOC measurements. It is also generally well written. However, some issues should be solved prior to publication.

SPECIFIC COMMENTS

1. The main issue, from my point of view, is the reliability of the measurements in the frame of the experimental setups. For example, the authors state that "The analysis of the internal lamp measurements in PTB1 shows a very marked nonlinear behaviour when using slit 5 and 6 relative to slit 2" (p. 9) and ascribe this behaviour to the internal halogen lamp. However, also the external lamp (slit 5-6) charts in Fig. 7 show some curvature above 40°C, which cannot be ascribed to the halogen lamp. Furthermore, looking at Fig. 6, I cannot understand the inconsistency of the results at PTB2 (internal lamp measurements show hysteresis, while external lamp measurements do not) and K&Z (vice-versa). It would be desiderable for the reader to have these issues explained better, in order to trust the results of the experiments (were the external lamps stable? Were temperatures measured reliably? Etc.)

2. I could have missed this information, but was the wavelength alignment ("hg tests") checked during the chamber experiments? It should be explained whether the final temperature sensitivity takes the wavelength shifts into account;

3. I cannot understand why tau_R6 (linear combination) is much more stable than the relative coefficients. Does this mean that F(306.3) is not a good reference? Or that noise is lower when combining the irradiances at 4 wavelengths compared to only 2 wavelengths?

4. It is stated that "The conclusions of this work cannot be extended to MkII and MkIV models" due to presence of NiSO4 filter. I agree with the authors that the temperature coefficients may vary between MkIII and other Brewer types, but why the main outcome of the paper (i.e. that the standard lamp can be effectively used to track the Brewer sensitivity to temperature changes) should be compromised?

5. Regarding the very last paragraph, recommending a change of the reference temperature, I am not sure whether this would reduce the uncertainty of the temperature correction. Indeed, since the correction is assessed based on experimental data, small measurement errors at ~22-23°C would result in lower deviations of the angular coefficient if the reference point is farther (0°C) from the reference. Instead, some issues could arise if the temperature dependence is locally linear about ~22-23°C, but globally not linear. In that case, I agree that changing the reference temperature would be a benefit.

6. Finally, according to the data usage rules of EUBREWNET, an acknowledgement to the PI's providing the data used in the paper (e.g., Fig. 1) should be included. I would suggest the authors to include the statement recommended on the EUBREWNET website: "We thank the European Brewer Network (http://rbcce.aemet.es/eubrewnet/) for providing access to the data and the PI investigators and their staff for establishing and maintaining the "#" sites used in this investigation".

TECHNICAL CORRECTIONS

- check usage of "internal lamp" vs more rigorous "internal halogen lamp" or "standard lamp" thoughout the paper. Indeed, two internal lamps are available in the Brewer (mercury and halogen);

- p. 1 line 18, "temperature-compensated" is not clear here, but the concept is explained in the following lines. Simply remove "temperature-compensated";

- p.2 line 28, "studied by different authors": please add bibliographic references;

- p. 3 line 20-24: rewrite this paragraph splitting the two points: 1) the weightings are chosen to minimise influence of SO2, linear effects and constant term; 2) the wavelengths are chosen to maximise sensitivity to ozone and to minimise small shifts in wavelengths (sun scan test);

- p. 4 Eq. 8: it is a common error. To comply with the Lambert-Beer-Bouguer equation, either Eq. 1 should read ETC – R6 or the cross section should be – sum(w_i alpha_i). Since the Brewer weightings give a negative differential cross section, it would look better if Eq. 8 had a "minus" sign;

- p. 6  line 1: "it" → "they";

- p. 6 line 17: define "Cte" (did you mean "constant"?)

- p. 6 line 18, "constant" → "constant over all wavelengths" (not in time);

- p. 7 line 17 and line 28: "Figure" → "Fig."

- p. 10 line 20: why the diurnal, and not the annual, variation was chosen to provide an idea of the internal temperature changes?

---

## Referee Report (RR3)

General comments:

The article provides a complete characterization of thermal sensitivity of the Brewer spectrophotometers in total ozone measurements. Although the topic addressed in the paper is very important for Brewer users, the issue of the temperature correction and the experimental procedure to investigate the temperature effect on ozone measurements can be also used for other instruments.

The paper is well-structured, all sections are well interrelated, and the objectives are clearly identified.

Specific comments:

Pag 1 L15: The authors should specify which kind of environmental parameters the instruments are exposed outdoors.

Pag 1 L20: which kind of changes are produced in the measured spectrum?

Pag 2 L4: How the internal temperature is measured should be specified here.

Pag2 L12: typo "approxiamation"

Pag 2 L15: for readers not familiar with TOC measurements with Brewer it needs to specify what is the "routine operation" and what are "the original coefficients".

Pag2 L28: Include some references about that issue.

Pag2 L 30: the acronym "EMRP ENV59" should be explained.

Pag 2 L 34: Specify the places of PTB (Physikalisch-Technische Bundesanstalt) and at Kipp &Zonen facilities

Fig1: specify at least in the legend what is the line inside the box, the top and bottom of each box are the 25th and 75th percentiles of the samples, and which values are set the whiskers.

Pag 7 L17: Typo "EUBRENET", include brackets to "(Figure 3)"

Pag. 7 L 28: why did the authors use only MKIII and not also MKII or MKIV which have shown higher $\tau R6$ than MKIII?

Pag 9 : typo in "this clearly observed behaviors"

Pag 9: Acknowledgements. As reported in Recommended guidelines for data use and publication of Eubrewnet data, the authors should write :" We thank the European Brewer Network (http://rbcce.aemet.es/eubrewnet/) for providing access to the data and the PI investigators and their staff for establishing and maintaining the "#" sites used in this investigation."

---

## Editor Decision (ED1)

February 28, 2018

Review of "Characterization of the instrument temperature dependence of Brewer total ozone column measurements" by Alberto Berjón et al.

Vladimir Savastiouk

The revised text of the paper addressed many of the concerns raised in the initial review and it reads much better now. I did find other areas where the paper can be improved, some of them are very important.

1. The title sounds as if the paper investigates the Brewer ozone calculation temperature dependence in general, but the content is mostly a comparison of 2 different experiments on 2 different Brewers plus a mentioning of variability of temperature coefficient values among the Brewers in EUBREWNET. I am not sure any general statements can be definitively drawn from this. A title similar to "An investigation of TC determination in EUBREWNET Brewers" can be more precise

P1 L20 Push rod doesn't really control the movement of the diffraction grating. Suggest not going into this details and say "materials used in the monochromator are selected to minimize the effect of the internal temperature changes on the spectrum position relative to the exit slits."

I will repeat my comment in the initial review: the paper does not clearly differentiate between two completely different temperature effects in the Brewer: the positioning of the spectrum and the changes in the spectral sensitivity. Some paragraphs refer to both effects that is extremely confusing. Moreover, the paper doesn't actually investigate anything relating to the spectral shifts and so it is unclear why the authors even go into this area. A simple statement that is in page 1 is more than enough and I feel that no mentioning of this should be anywhere in the paper after that.

P2 L5 "During the test, the diffraction grating is positioned such that the operating wavelengths are dispersed onto the appropriate exit slits" - please rephrase as this sentence doesn't describe much.

P2 L14 TC are calculated for all operating wavelengths, not just those used for ozone

P2 L25 replace "the" with "some" in "quartz window and the neutral density filters"

P3 L7 remove double "an"

P3 L15 it's not really an "alternative", it's the same

P3 L19 I refuse to accept your explanation that since you've already used this notation for the slit numbering in another paper and this makes it ok to use it again. It is wrong to use unconventional terminology, Many papers before yours used it correctly.

P6 L10-18  Should not be in this paper.

P7 L18  Have you looked at the fact that Brewers have at least three different PMT types and that can contribute to the differences in the TC? If you did, what was the result? If not, why not?

P7 L22,25 The order filter in not just $NiSO_4$ crystal, but a combination filter with two UG11 glass filters.

P10 L20 *** An important point!!! *** : did you calculate the slopes and their uncertainties using the averages for each temperature? It very much looks like you did and then your uncertainty is incorrect as you forgot that each point (each average) has an uncertainty associated with it already (a very large standard deviation in fact).  You should re-calculate those uncertainties using each point or propagate the uncertainties of the averages to that of the slopes. When you do this you will likely find that the uncertainty is larger than 100% for most of your slopes, which brings back my original point from the initial review: if  for each temperature you have a spread of R6 so large that is close to the spread between R6 at extreme temperatures you cannot actually correct or improve such data. Whatever you do you will still have that spread. So, the question then is can you even trust such a data for determining TC?

P11 L1 I am not sure I understand why you are saying "in spite of robustness of the TOC calculation algorithm". The algorithm clearly works. It was the instrument (the hardware) that didn't.

 P11 L5 You may consider stating clearly that it is imperative to schedule SL tests throughout the day to cover the different temperatures inside the Brewer.

GENERAL COMMENTS

This paper addresses an important issue for the Brewer users and ozone communities, since an accurate assessment of the Brewer temperature dependence is essential to ensure reliable TOC measurements. It is also generally well written. However, some issues should be solved prior to publication.

SPECIFIC COMMENTS

1. The main issue, from my point of view, is the reliability of the measurements in the frame of the experimental setups. For example, the authors state that "The analysis of the internal lamp measurements in PTB1 shows a very marked nonlinear behaviour when using slit 5 and 6 relative to slit 2" (p. 9) and ascribe this behaviour to the internal halogen lamp. However, also the external lamp (slit 5-6) charts in Fig. 7 show some curvature above 40°C, which cannot be ascribed to the halogen lamp. Furthermore, looking at Fig. 6, I cannot understand the inconsistency of the results at PTB2 (internal lamp measurements show hysteresis, while external lamp measurements do not) and K&Z (vice-versa). It would be desiderable for the reader to have these issues explained better, in order to trust the results of the experiments (were the external lamps stable? Were temperatures measured reliably? Etc.)

2. I could have missed this information, but was the wavelength alignment ("hg tests") checked during the chamber experiments? It should be explained whether the final temperature sensitivity takes the wavelength shifts into account;

3. I cannot understand why tau_R6 (linear combination) is much more stable than the relative coefficients. Does this mean that F(306.3) is not a good reference? Or that noise is lower when combining the irradiances at 4 wavelengths compared to only 2 wavelengths?

4. It is stated that "The conclusions of this work cannot be extended to MkII and MkIV models" due to presence of NiSO4 filter. I agree with the authors that the temperature coefficients may vary between MkIII and other Brewer types, but why the main outcome of the paper (i.e. that the standard lamp can be effectively used to track the Brewer sensitivity to temperature changes) should be compromised?

5. Regarding the very last paragraph, recommending a change of the reference temperature, I am not sure whether this would reduce the uncertainty of the temperature correction. Indeed, since the correction is assessed based on experimental data, small measurement errors at ~22-23°C would result in lower deviations of the angular coefficient if the reference point is farther (0°C) from the reference. Instead, some issues could arise if the temperature dependence is locally linear about ~22-23°C, but globally not linear. In that case, I agree that changing the reference temperature would be a benefit.

6. Finally, according to the data usage rules of EUBREWNET, an acknowledgement to the PI's providing the data used in the paper (e.g., Fig. 1) should be included. I would suggest the authors to include the statement recommended on the EUBREWNET website: "We thank the European Brewer Network (http://rbcce.aemet.es/eubrewnet/) for providing access to the data and the PI investigators and their staff for establishing and maintaining the "#" sites used in this investigation".

TECHNICAL CORRECTIONS

- check usage of "internal lamp" vs more rigorous "internal halogen lamp" or "standard lamp" thoughout the paper. Indeed, two internal lamps are available in the Brewer (mercury and halogen);

- p. 1 line 18, "temperature-compensated" is not clear here, but the concept is explained in the following lines. Simply remove "temperature-compensated";

- p.2 line 28, "studied by different authors": please add bibliographic references;

- p. 3 line 20-24: rewrite this paragraph splitting the two points: 1) the weightings are chosen to minimise influence of SO2, linear effects and constant term; 2) the wavelengths are chosen to maximise sensitivity to ozone and to minimise small shifts in wavelengths (sun scan test);

- p. 4 Eq. 8: it is a common error. To comply with the Lambert-Beer-Bouguer equation, either Eq. 1 should read ETC – R6 or the cross section should be – sum(w_i alpha_i). Since the Brewer weightings give a negative differential cross section, it would look better if Eq. 8 had a "minus" sign;

- p. 6  line 1: "it" → "they";

- p. 6 line 17: define "Cte" (did you mean "constant"?)

- p. 6 line 18, "constant" → "constant over all wavelengths" (not in time);

- p. 7 line 17 and line 28: "Figure" → "Fig."

- p. 10 line 20: why the diurnal, and not the annual, variation was chosen to provide an idea of the internal temperature changes?

General comments:

The article provides a complete characterization of thermal sensitivity of the Brewer spectrophotometers in total ozone measurements. Although the topic addressed in the paper is very important for Brewer users, the issue of the temperature correction and the experimental procedure to investigate the temperature effect on ozone measurements can be also used for other instruments.

The paper is well-structured, all sections are well interrelated, and the objectives are clearly identified.

Specific comments:

Pag 1 L15: The authors should specify which kind of environmental parameters the instruments are exposed outdoors.

Pag 1 L20: which kind of changes are produced in the measured spectrum?

Pag 2 L4: How the internal temperature is measured should be specified here.

Pag2 L12: typo "approxiamation"

Pag 2 L15: for readers not familiar with TOC measurements with Brewer it needs to specify what is the "routine operation" and what are "the original coefficients".

Pag2 L28: Include some references about that issue.

Pag2 L 30: the acronym "EMRP ENV59" should be explained.

Pag 2 L 34: Specify the places of PTB (Physikalisch-Technische Bundesanstalt) and at Kipp &Zonen facilities

Fig1: specify at least in the legend what is the line inside the box, the top and bottom of each box are the 25th and 75th percentiles of the samples, and which values are set the whiskers.

Pag 7 L17: Typo "EUBRENET", include brackets to "(Figure 3)"

Pag. 7 L 28: why did the authors use only MKIII and not also MKII or MKIV which have shown higher $\tau R6$ than MKIII?

Pag 9 : typo in "this clearly observed behaviors"

Pag 9: Acknowledgements. As reported in Recommended guidelines for data use and publication of Eubrewnet data, the authors should write :" We thank the European Brewer Network (http://rbcce.aemet.es/eubrewnet/) for providing access to the data and the PI investigators and their staff for establishing and maintaining the "#" sites used in this investigation."

---

## Author Response (AR2)

**List of changes on "Study on the effect of instrumental temperature on Brewer total ozone column measurements".**

Alberto Berjón et al.

We would like to thank the three referees for the suggestions which definitely help to improve this paper. We include below our point-by-point response to the referees comments and a version of the paper with all changes highlighted.

**Response to Vladimir Savastiouk**

The revised text of the paper addressed many of the concerns raised in the initial review and it reads much better now. I did find other areas where the paper can be improved, some of them are very important.

1. The title sounds as if the paper investigates the Brewer ozone calculation temperature dependence in general, but the content is mostly a comparison of 2 different experiments on 2 different Brewers plus a mentioning of variability of temperature coefficient values among the Brewers in EUBREWNET. I am not sure any general statements can be definitively drawn from this. A title similar to "An investigation of TC determination in EUBREWNET Brewers" can be more precise,

> **As shown in the results, the temperature coefficients ($\tau_b$ and $\tau'_b$) can be very different depending on the method used to obtain it, but the effect on ozone ($\tau_{R6}$) does not depend on the used method. We think this result can be extended to other instruments since it seems to be due to the linear combination used in the Brewer algorithm. Therefore the title proposed by the referee does not meet the aims of this work. Nevertheless, in order to clarify the object of the study, the title of the work has been modified as: "Study on the effect of instrumental temperature on Brewer total ozone column measurements"**

P1 L20 Push rod doesn't really control the movement of the diffraction grating. Suggest not going into this details and say "materials used in the monochromator are selected to minimize the effect of the internal temperature changes on the spectrum position relative to the exit slits."

> **We have replaced the sentence suggested by the referee.**

I will repeat my comment in the initial review: the paper does not clearly differentiate between two completely different temperature effects in the Brewer: the positioning of the spectrum and the changes in the spectral sensitivity. Some paragraphs refer to both effects that is extremely confusing. Moreover, the paper doesn't actually investigate anything relating to the spectral shifts and so it is unclear why the authors even go into this area. A simple statement that is in page 1 is more than enough and I feel that no mentioning of this should be anywhere in the paper after that.

> **To clarify this point we have modify the second and third paragraphs of the introduction. The main idea is to introduce the effects by which temperature can modify the Brewer measurements. But as changes in wavelength selection are of special relevance for the ozone determination, we introduce the mechanism already used in the brewer to avoid this effect.**

P2 L5 "During the test, the diffraction grating is positioned such that the operating wavelengths are dispersed onto the appropriate exit slits" - please rephrase as this sentence doesn't describe much.

**We think this sort description of the HG test is enough for the scope of this paper. It is slightly modified from the "BREWER COMMAND SUMMARY" (pag. 95) of the "MKIII OPERATOR'S MANUAL" (http://www.kippzonen.com/Download/207/Brewer-MkIII-Operator-s-Manual). We include this reference in the revised manuscript for those interested in more details about this test.**

P2 L14 TC are calculated for all operating wavelengths, not just those used for ozone

**The sentence is modified as "Temperature coefficients in a linear approximation are determined during this characterization for all Brewer operating wavelengths."**

P2 L25 replace "the" with "some" in "quartz window and the neutral density filters"

**This is replaced.**

P3 L7 remove double "an"

**This is removed.**

P3 L15 it's not really an "alternative", it's the same

**"but can be alternatively defined" is modified as "but can be also written"**

P3 L19 I refuse to accept your explanation that since you've already used this notation for the slit numbering in another paper and this makes it ok to use it again. It is wrong to use unconventional terminology, Many papers before yours used it correctly.

**We agree with the referee that the numbering assigned to the slit may not be the standard even if it can be found widely in the bibliography. We change the nomenclature to slit2 -> 310.0 nm, slit3 -> 313.5 nm, slit4 -> 316.8 nm and slit5 -> 320.0 nm.**

P6 L10-18 Should not be in this paper.

**This section is removed.**

P7 L18 Have you looked at the fact that Brewers have at least three different PMT types and that can contribute to the differences in the TC? If you did, what was the result? If not, why not?

**This is one of the reasons we have carried out a statistical analysis of the temperature sensitivity of the different Brewer models. However, we have not seen significant differences beyond the differences shown between the models.**

P7 L22,25 The order filter in not just NiSO4 crystal, but a combination filter with two UG11 glass filters.

**We rename "NiSO$_4$ filter" by "NiSO$_4$/UG11 filter"**

P10 L20 *** An important point!!! *** : did you calculate the slopes and their uncertainties using the averages for each temperature? It very much looks like you did and then your uncertainty is incorrect as you forgot that each point (each average) has an uncertainty associated with it already (a very large standard deviation in fact). You should re-calculate those uncertainties using each point or propagate

the uncertainties of the averages to that of the slopes. When you do this you will likely find that the uncertainty is larger than 100% for most of your slopes, which brings back my original point from the initial review: if for each temperature you have a spread of R6 so large that is close to the spread between R6 at extreme temperatures you cannot actually correct or improve such data. Whatever you do you will still have that spread. So, the question then is can you even trust such a data for determining TC?

**We are very confident in the TC shown in the present work. Even if not all the data variation is explained by the temperature, as it is clearly shown by the $R^2$ values, the dependence of the data with the temperature is clearly shown in the plots.**

**The linear regressions were done using the averages for each temperature. This was done to prevent a overrepresentation of those temperatures with a higher number of measurements. Nevertheless, in order to avoid any doubt, we have now included a second analysis with the results when linear regressions are done using individual measurements. The results show no strong differences between the standard error from both methods, as it depends on the number of measurements we are using in the linear regression. But in a few cases, some overrepresentation is detected when using individual values. All these considerations are included in the manuscript.**

P11 L1 I am not sure I understand why you are saying "in spite of robustness of the TOC calculation algorithm". The algorithm clearly works. It was the instrument (the hardware) that didn't.

**We try to emphasize that it is not possible to obtain correct results if the internal halogen lamp varies greatly. Nevertheless, the sentence is removed from the revised manuscript to avoid any confusion.**

P11 L5 You may consider stating clearly that it is imperative to schedule SL tests throughout the day to cover the different temperatures inside the Brewer.

**We agree with the referee and we included in the Conclusions section the following sentence: "In order to apply this method it is advisable to schedule SL tests throughout the day to record as wide a range of temperature variation as possible."**

**Response to Referee #3**

GENERAL COMMENTS

This paper addresses an important issue for the Brewer users and ozone communities, since an accurate assessment of the Brewer temperature dependence is essential to ensure reliable TOC measurements. It is also generally well written. However, some issues should be solved prior to publication.

SPECIFIC COMMENTS

1. The main issue, from my point of view, is the reliability of the measurements in the frame of the experimental setups. For example, the authors state that "The analysis of the internal lamp measurements in PTB1 shows a very marked nonlinear behaviour when using slit 5 and 6 relative to slit 2" (p. 9) and ascribe this behaviour to the internal halogen lamp. However, also the external lamp (slit 5-6) charts in Fig. 7 show some curvature above 40°C, which cannot be ascribed to the halogen lamp. Furthermore, looking at Fig. 6, I cannot understand the inconsistency of the results at PTB2 (internal lamp measurements show hysteresis, while external lamp measurements do not) and K&Z (vice-versa). It would be desirable for the reader to have these issues explained better, in order to trust the results of the experiments (were the external lamps stable? Were temperatures measured reliably? Etc.)

> **Two main issues considered as responsibles of the failed result in the first experiment at the PTB are the halogen internal lamp and the alignment of the external light source with the direct entrance port. As described in the text , in PTB1 "A Hamamatsu model LC8 UV source with a built-in Xe lamp and equipped with a quartz fiber bundle as a light guide was used to illuminate simultaneously both global and direct input ports of the Brewer". But in PTB2 "On this second occasion, referred to as PTB2, the external lamp was aligned with the direct entrance port.". We have rewritten Results and Conclusions sections to reinforced this idea.**

> **The inconsistencies shown in Figure 6, in spite of using a stabilized external lamp, as in the K&Z experiment, or correcting the observed variations through monitors, as in the case of PTB experiments, reflect the difficulty of determining absolute temperature coefficients. In Figure 6 it should be noted that the total variation of the measurements F(310.1nm) is around 1% when temperature change more than 50 °C. One of the main points of this work is to show how the combination of the measurements at different wavelengths, specially when combining the four channels, reduces the possible uncontrolled effects and allow us to determine the temperature sensitivity of the Brewer. The results and conclusions are rewritten to clearly show this point.**

2. I could have missed this information, but was the wavelength alignment ("hg tests") checked during the chamber experiments? It should be explained whether the final temperature sensitivity takes the wavelength shifts into account;

> **The standard procedure to correct wavelength shifts was applied in every cycle. We write in the manuscript: "In all cases, each measurement cycle included an HG test, repositioning the micrometer of the diffraction grating to locate the 302.15nm line of the mercury discharge lamp."**

3. I cannot understand why tau_R6 (linear combination) is much more stable than the relative coefficients. Does this mean that F(306.3) is not a good reference? Or that noise is lower when combining the irradiances at 4 wavelengths compared to only 2 wavelengths?

> **The combination of the four wavelengths clearly increase the stability in both experiments. When combining the irradiances at 4 wavelengths any linear effects with wavelength is suppressed (Equation 5). But this not happened when combining only 2 wavelengths. We clarify this point in the Conclusions as follows:**

> **"The calculated $\tau_{R6}$ are very much stable. The combination of the four wavelengths clearly increase the stability in all the experiments. This is probably because the linear combination**

**removes any linear effect with the wavelength, as it verify Equation 5. Instead, the relative coefficients does not verify this property."**

4. It is stated that "The conclusions of this work cannot be extended to MkII and MkIV models" due to presence of NiSO4 filter. I agree with the authors that the temperature coefficients may vary between MkIII and other Brewer types, but why the main outcome of the paper (i.e. that the standard lamp can be effectively used to track the Brewer sensitivity to temperature changes) should be compromised?

> **The main source of thermal sensitivity in the MKII and MKIV models seems to be due to the NiSO4 filter. Therefore, we prefer no to generalize a result that has not been validated in our experiment. We rewrite the sentence to clarify this point:**
>
> **"Therefore, the conclusions of this work may not be directly applicable to the MKII and MKIV models. Further studies are necessary in order to analyze these specific models."**

5. Regarding the very last paragraph, recommending a change of the reference temperature, I am not sure whether this would reduce the uncertainty of the temperature correction. Indeed, since the correction is assessed based on experimental data, small measurement errors at ~22-23°C would result in lower deviations of the angular coefficient if the reference point is farther (0°C) from the reference. Instead, some issues could arise if the temperature dependence is locally linear about ~22-23°C, but globally not linear. In that case, I agree that changing the reference temperature would be a benefit.

> **The reduction in uncertainty should be understood in the total data set when considering the uncertainty of the temperature coefficient. The majority of the data of the different Brewer are measured at an internal temperature close to 22 °C, but very few are measured near 0 °C. At the end of the Results section we now include a study of how such a change could affect the uncertainty derived for the data set in the EUBREWNET network.**

6. Finally, according to the data usage rules of EUBREWNET, an acknowledgement to the PI's providing the data used in the paper (e.g., Fig. 1) should be included. I would suggest the authors to include the statement recommended on the EUBREWNET website: "We thank the European Brewer Network (http://rbcce.aemet.es/eubrewnet/) for providing access to the data and the PI investigators and their staff for establishing and maintaining the "#" sites used in this investigation".

> **We modify the acknowledgments to include the Eubrewnet recommendation.**
>
> **"We thank the European Brewer Network (http://rbcce.aemet.es/eubrewnet/) for providing access to the data and the PI investigators and their staff for establishing and maintaining the 32 sites used in this investigation. We further acknowledge the support of the Fundación General de la Universidad de La Laguna."**

TECHNICAL CORRECTIONS

- check usage of "internal lamp" vs more rigorous "internal halogen lamp" or "standard lamp" thoughout the paper. Indeed, two internal lamps are available in the Brewer (mercury and halogen);

> **"internal lamp" is replaced by "internal halogen lamp"**

- p. 1 line 18, "temperature-compensated" is not clear here, but the concept is explained in the following lines. Simply remove "temperature-compensated";

> **We will keep the term "temperature-compensated" in the text as is the term used in the cited reference (McElroy, 2014). We think this term faithfully describe the monochromator designed for the Brewer, and as the concept is explained in the following lines it should be clear enough.**

- p.2 line 28, "studied by different authors": please add bibliographic references;

**Three relevant references about that issue are now included: (Weatherhead et al., 2001; Siani et al., 2003; Fountoulakis et al., 2017).**

- p. 3 line 20-24: rewrite this paragraph splitting the two points: 1) the weightings are chosen to minimise influence of SO2, linear effects and constant term; 2) the wavelengths are chosen to maximise sensitivity to ozone and to minimise small shifts in wavelengths (sun scan test);

**The sentence is rewritten to clearly split the two points:**

**"The wavelengths, λi, used in Equations 2 and 3 have been especially selected to minimize any small shift in wavelength (Fioletov et al., 2005). Moreover, coefficients ωi have been determined to suppress any influence of the aerosol and the SO2 in the ozone retrieval (Dobson, 1957; Kerr et al., 1981). In general any linear effects with wavelength is also suppressed, as λi and ωi verify Equations 4 and 5."**

- p. 4 Eq. 8: it is a common error. To comply with the Lambert-Beer-Bouguer equation, either Eq. 1 should read ETC – R6 or the cross section should be – sum(w_i alpha_i). Since the Brewer weightings give a negative differential cross section, it would look better if Eq. 8 had a "minus" sign;

**We correct the error in Equation 1 writing ETC-R6.**

- p. 6 line 1: "it" → "they";

**The typo is corrected.**

- p. 6 line 17: define "Cte" (did you mean "constant"?)

**We replaced "Cte" by "constant"**

- p. 6 line 18, "constant" → "constant over all wavelengths" (not in time);

**We replaced "constant" by "constant over all wavelengths"**

- p. 7 line 17 and line 28: "Figure" → "Fig."

**We keep "Figure" in these two cases as it is the term used all along the text.**

- p. 10 line 20: why the diurnal, and not the annual, variation was chosen to provide an idea of the internal temperature changes?

**We clarify this point with the following sentence:**
**"As the mean diurnal variation is close to 10°C, that value can be considered the diurnal uncertainty due to the temperature correction. This result is important when we try to distinguish between different operating issues that may generate diurnal cycles, as wrong temperature coefficients or incorrect values of ETC."**

**Response to Referee #4**

General comments:

The article provides a complete characterization of thermal sensitivity of the Brewer spectrophotometers in total ozone measurements. Although the topic addressed in the paper is very important for Brewer users, the issue of the temperature correction and the experimental procedure to investigate the temperature effect on ozone measurements can be also used for other instruments.

The paper is well-structured, all sections are well interrelated, and the objectives are clearly identified.

Specific comments:

Pag 1 L15: The authors should specify which kind of environmental parameters the instruments are exposed outdoors.

>**The present work focuses on the analysis of the effect of temperature on measurements, and therefore at this point we refer exclusively to temperature. To avoid ambiguities, the paragraph is rewritten as follows:**
>
>**"To be able to make this measurements, this equipment should be suitable for outdoor use. Moreover, this instrument should operate at any temperature, since it is installed on a wide range of environments, from subtropical deserts to polar zones."**

Pag 1 L20: which kind of changes are produced in the measured spectrum?

>**The paragraph has been rewritten to address the comments of other referees, but we have include the following sentence that may response to this question:**
>
>**"Changes in temperature may affect the Brewer in two different ways: changing its sensitivity and causing a spectrum shift."**

Pag 2 L4: How the internal temperature is measured should be specified here.

>**The sentence describing how the internal temperature is measured is moved from Pag 7 to Pag 2, and the sentence is rewritten as:**
>
>**"Thus the Brewer operational procedure recommends to perform an internal Hg-lamp test (HG test) when the internal temperature, which is registered for each ozone measurement by a sensor located near the PMT, varies more than 3°"**

Pag2 L12: typo "approxiamation"

>**The typo is corrected.**

Pag 2 L15: for readers not familiar with TOC measurements with Brewer it needs to specify what is the "routine operation" and what are "the original coefficients".

>**The sentence is rewritten to clarify the procedure of determining the temperature coefficients during the calibration campaigns:**
>
>**"If an appreciable temperature dependence in the retrieved ozone is detected during the calibration campaigns, the temperature coefficients are corrected using the in-field data of the internal halogen lamp measurements along the diurnal temperature variation (Redondas and Rodríguez-Franco, 2015)."**

Pag2 L28: Include some references about that issue.

**Three relevant references about that issue are now included: (Weatherhead et al., 2001; Siani et al., 2003; Fountoulakis et al., 2017).**

Pag2 L 30: the acronym "EMRP ENV59" should be explained.

**The acronym EMRP (European Metrology Research Programme) is now explained in the text. The project identification (ENV59) is removed as the project is clearly identified by the tittle. The sentence is rewritten as:**

**"On this basis, the validation of the procedures for the retrieval of the temperature coefficients was included as one of the objectives of the "Traceability for atmospheric total column ozone" (ATMOZ) project of the European Metrology Research Programme (EMRP)."**

Pag 2 L 34: Specify the places of PTB (Physikalisch-Technische Bundesanstalt) and at Kipp &Zonen facilities

**The places of both facilities are now included and the sentence is rewritten as:**

**"For this purpose, we have made measurements with #185 and #233 MKIII Brewer spectrophotometers respectively at PTB (Physikalisch-Technische Bundesanstalt) in Braunschweig, Germany, and Kipp &Zonen in Delft, Netherlands."**

Fig1: specify at least in the legend what is the line inside the box, the top and bottom of each box are the 25th and 75th percentiles of the samples, and which values are set the whiskers.

**A description of the different elements is included in the figure legend as follows:**

**"Median temperatures are represented by lines within boxes determined by the 1st and 3rd quartiles. Whiskers represent Tukey's limits."**

Pag 7 L17: Typo "EUBRENET", include brackets to "(Figure 3)"

**The typo is corrected.**

Pag. 7 L 28: why did the authors use only MKIII and not also MKII or MKIV which have shown higher τR6 than MKIII?

**We have included the following sentence in the Conclusions to clarify this point:**
**"We have prioritized the MKIII model in this study since it is the most extended model and generally used as reference, as in the case of the RBCC-E."**

Pag 9 : typo in "this clearly observed behaviors"

**The typo is corrected.**

Pag 9: Acknowledgements. As reported in Recommended guidelines for data use and publication of

[revised manuscript text omitted]

---

## Author Response (AR3)

**List of changes on "Sensitivity study of the instrumental temperature corrections on Brewer total ozone column measurements".**

Alberto Berjón et al.

We would like to thank the editor for her time spent on reviewing our manuscript and her comments helping us improving the article.

The English structure and grammar of the manuscript has been thoroughly reviewed. The title of the manuscript has been changed to "Sensitivity study of the instrumental temperature corrections on Brewer total ozone column measurements" .

We include below our point-by-point response to the editor comments and a version of the paper with all changes highlighted.

**Associate Editor Decision: Publish subject to technical corrections** (16 May 2018) by Irina Petropavlovskikh

Comments to the Author:

Dear Authors,
Thank you for submitting responses to comments from three reviewers. The discussion of results was significantly improved. Based on modifications to the manuscript, I made a decision to accept the paper for publication.

The manuscript text still has many grammatical errors.

For example in lines 8 and 9 of the abstract, the following sentence was edited, but now the meaning of what you are trying to say is lost.
"The results clearly show that the traditional methodology based on the internal halogen lamps is not affected by possible temperature affecting the light source". Is it better to say "The results clearly show that the traditional methodology based on the internal halogen lamps is not sensitive to the temperature-caused changes in the spectrum of the light source"? Is it external or internal light source? Please add to. the sentence

> **The sentence is replaced by:**
> **"The results clearly show that the traditional methodology based on the internal halogen lamps is not sensitive to the temperature-caused changes in the spectrum of the internal light source."**

P.1 ,line 17, it is installed "IN" the wide range…

> **This is replaced.**

P. 2, line 3, suggestion to change to "Preventing mistakes in the wavelength selection are of importance...."

> **The sentence is replaced by:**
> **"Preventing errors in the wavelength selection is of highest importance for the ozone determination."**

p. 2 line 6 "(?)" - was it a placeholder for the reference to a paper?
p. 2 line 13, (?) - please remove.
There are several places in the text when the sentence ends with "(?)". Please remove or add reference if it is missing.

**Issues with references have been corrected.**

p. 6, line 19 Not sure what you mean by "verify Equation (5)" Do you mean "confirm to the condition described by Equation (5)"?

**The sentence is replaced by:**
**"satisfy the conditions defined by Equation"**

p. 7 line 9 suggestion to change to "...for the experiments, sensitivity study was performed."

**The sentence is replaced by:**
**"In order to determine the most suitable temperature range for the experiments, a statistical analysis of the Brewer operating temperatures using the EUBREWNET database was performed."**

p. 10, line 19 "relative" -> "relative coefficient "? Not clear what you are saying here. Please clarify.

**The sentence is replaced by:**
**"We can clearly see the improvement when using the analysis of the relative coefficients. In most of the cases we can assume a linear relationship between relative Brewer measurements and the temperatures."**

p. 10 , line 22 - "are" - > "is" , i.e. "the summary is shown"

**The sentence is replaced by:**
**"A summary of the results obtained by both linear regressions methods is shown in Tables 1 and 2."**

p. 11, line 17, "show" -> "found"

**This is replaced.**

p. 11, lines 22-23. This sentence is unfinished.

**The sentence is replaced by:**
**"This result might be useful when analyzing different operating issues, such as wrong temperature coefficients or incorrect values of ETC, that may introduce diurnal cycles in the final ozone values."**

p. 13, line 7-8, another use of "verify Equation (5)". May be "satisfies the conditions defined by Equation (5)"? Please re-write this and next sentence.

**This is replaced.**

p. 13, line 9 -should be "is done"

**This is replaced.**

Please read the manuscript carefully after finalizing the edits and correct text with proper English. You might need to find the native English speaker to proof read your manuscript before the re-submission of the manuscript.

**The English structure and grammar of the manuscript has been thoroughly reviewed.**

The paper should be resubmitted after the technical corrections are done.

Thank you for the valuable contribution to the special issue.
Regards,
Irina Petropavlovskikh

[revised manuscript text omitted]